# Support vector machines and linear regression coincide with very high-dimensional features

**Navid Ardeshir***
Dept. of Statistics
Columbia University
na2844@columbia.edu

**Clayton Sanford***
Dept. of Computer Science
Columbia University
clayton@cs.columbia.edu

**Daniel Hsu**
Dept. of Computer Science
Columbia University
djhsu@cs.columbia.edu

## Abstract

The support vector machine (SVM) and minimum Euclidean norm least squares regression are two fundamentally different approaches to fitting linear models, but they have recently been connected in models for very high-dimensional data through a phenomenon of support vector proliferation, where every training example used to fit an SVM becomes a support vector. In this paper, we explore the generality of this phenomenon and make the following contributions. First, we prove a super-linear lower bound on the dimension (in terms of sample size) required for support vector proliferation in independent feature models, matching the upper bounds from previous works. We further identify a sharp phase transition in Gaussian feature models, bound the width of this transition, and give experimental support for its universality. Finally, we hypothesize that this phase transition occurs only in much higher-dimensional settings in the $\ell_1$ variant of the SVM, and we present a new geometric characterization of the problem that may elucidate this phenomenon for the general $\ell_p$ case.

## 1 Introduction

The *support vector machine* (SVM) and *ordinary least squares* (OLS) are well-weathered approaches to fitting linear models, but they are associated with different learning tasks: classification and regression. In this paper, we study the case in which the models return exactly the same hypothesis for sufficiently high-dimensional data.

The hard-margin SVM is a linear classification model that finds the separating hyperplane that maximizes the minimum margin of error for every training sample. If the training data $(\mathbf{x}_1, y_1), \ldots, (\mathbf{x}_n, y_n) \in \mathbb{R}^d \times \{\pm 1\}$ are linearly separable, then the resulting linear classifier is $x \mapsto \text{sign}\left(x^\intercal w_{\text{SVM}}\right)$, where $w_{\text{SVM}}$ is the solution to the following optimization problem:

$$w_{\text{SVM}} = \arg\min_{w \in \mathbb{R}^d} \|w\|_2 \quad \text{such that} \quad y_i w^\intercal \mathbf{x}_i \geq 1, \ \forall i \in [n]. \tag{1}$$

An example $\mathbf{x}_i$ is a *support vector* if the corresponding constraint is satisfied with equality, and the optimal solution $w_{\text{SVM}}$ is a linear combination of these support vectors.

Ordinary least squares regression finds the linear function that best fits the training data $(\mathbf{x}_1, y_1), \ldots, (\mathbf{x}_n, y_n) \in \mathbb{R}^d \times \mathbb{R}$ according to the sum of squared errors. When the solution is not unique, it is natural to take the solution of minimum Euclidean norm; this is the convention we adopt. Taking $\mathbf{X} := [\mathbf{x}_1 | \ldots | \mathbf{x}_n]^\intercal \in \mathbb{R}^{n \times d}$ and $y := (y_1, \ldots, y_n)$, the solution is the hypothesis $x \mapsto w_{\text{OLS}}^\intercal x$ where $w_{\text{OLS}}$ is the solution to the following: $w_{\text{OLS}} = \arg\min_{w \in \mathbb{R}^d} \|w\|_2$ such that $\mathbf{X}^\intercal \mathbf{X} w = \mathbf{X}^\intercal y$. In many high-dimensional settings (e.g., where $\mathbf{X}$ has full row rank), the solution may in fact *interpolate* the training data, i.e.,

$$w_{\text{OLS}} = \arg\min_{w \in \mathbb{R}^d} \|w\|_2 \quad \text{such that} \quad w^\intercal \mathbf{x}_i = y_i, \ \forall i \in [n]. \tag{2}$$

35th Conference on Neural Information Processing Systems (NeurIPS 2021).

Although the optimization problems in (1) and (2) are very different, they have been observed to coincide in very high-dimensional regimes. The study of this *support vector proliferation* (SVP) phenomenon—in which every training example is a support vector—was recently initiated by Muthukumar et al. [35] and Hsu et al. [23]. Roughly speaking, they show that SVP occurs when $d = \Omega(n \log n)$ for a broad class of sample distributions, and that SVP does not occur when $d = O(n)$ in an idealized isotropic Gaussian case.

SVP is a phenomenon that connects linear classification and linear regression, topics that have received renewed attention due to the break-down of classical analyses of these methods in high-dimensions. For instance, some analyses of SVM that are based on the number of support vectors become vacuous when this number becomes large [e.g., 19, 20, 44]. Similarly, overparameterized linear regression is typically only studied in noisy settings with explicit regularization. It was not until recently that SVM and OLS have been meaningfully analyzed in these regimes (see Section 1.2), and the connection between the two approaches via SVP has played an important analytical role [9, 35, 46].

In this work, we further examine support vector proliferation with the goal of broadly understanding when and why SVMs and OLS coincide. We pose and study the following questions:

1. *How general is the SVP phenomena? What relationship between $d$ and $n$ determines if the solutions to* (1) *and* (2) *coincide?*

   We close the $\log n$ gap from the prior work of Hsu et al. [23] by showing that $d \gtrsim n \log n$ is *necessary* for SVP to occur under a model of independent subgaussian features, even with constant probability. Our lower-bounds hold for a broad class of distributions over $\mathbf{x}_i$, and they match the upper-bounds from [23]. This demonstrates that SVP is extremely unlikely to occur in the much-studied $d = \Theta(n)$ setting.

2. *Is there a sharp threshold separating the occurrence and non-occurrence of this phenomenon? Is this threshold universal across all "reasonable" distributions over each $\mathbf{x}_i$?*

   We hypothesize that a sharp phase transition occurs at $d = 2n \log n$. We rigorously prove this hypothesis for isotropic Gaussian features and quantitatively bound the width of the transition. We experimentally observe the same transition for a wide range of other distributions.

3. *Is support vector proliferation specific to the $\ell_2$ SVM problem? If* (1) *and* (2) *are generalized to instead minimize $\ell_p$ norms, does this still occur at the same rate?*

   We re-frame this question with a geometric characterization of the dual of the SVM optimization problem for $\ell_p$ norms. We conjecture that a similar phase transition occurs for $\ell_1$, but also that it requires much larger dimension $d$; this is supported by preliminary experiments.

## 1.1  Outline of our results

Section 2 introduces the SVM and OLS approaches in full generality, our $\lambda$-anisotropic subgaussian data model, and prior results about SVP. Several equivalent characterizations of SVP are established (Proposition 1) for use in subsequent sections.

Section 3 characterizes when SVP *does not* occur for a broad range of distributions (Theorem 3). Our lower-bound on the dimension required for SVP matches the upper-bounds from [23] in the isotropic Gaussian setting, resolving the open question from that work, and also gives new lower-bounds for anisotropic cases. The proof works by tightly controlling the spectrum of the Gram matrix and establishing anti-concentration via the Berry-Esseen Theorem.

Section 4 establishes a sharp threshold of $d = 2n \log n$ for SVP in the case of isotropic Gaussian samples, and also characterize the width of the phase transition (Theorem 4).

Section 5 provides empirical evidence that the sharp threshold observed in Section 4 holds for a wide range of random variables. Rigorous statistical methodology inspired by Donoho and Tanner [16] is used to test our "universality hypothesis" that the probability of SVP does not depend on the underlying sample distribution as $d$ and $n$ become large.

Section 6 asks the questions about SVP from the preceding sections in the context of $\ell_1$-SVM and minimum $\ell_1$-norm interpolation. Specifically, the SVP threshold for $\ell_1$ is conjectured to occur for $d = \omega(n \log n)$. Evidence for this conjecture is provided in a simulation study and in geometric arguments about random linear programs.

## 1.2 Related work

**Prior works connecting SVP and generalization.** Muthukumar et al. [35] initiate the study of SVP in part to facilitate generalization analysis of the SVM in very high-dimensional settings. Their work, as well as the contemporaneous work of Chatterji and Long [10], shows that the SVM enjoys low test error in certain regimes where classical learning-theoretic analyses would otherwise yield vacuous error bounds. (In fact, one of the settings in [10] requires polynomially-higher dimension than is typically studied: $d = \Omega(n^2 \log n)$.) The coincidence between SVM and OLS identified by Muthukumar et al. was also more recently used by Wang and Thrampoulidis [46] and Cao et al. [9] for analyses of linear classification in very high dimensions under different data distributions.

The generalization analysis of Muthukumar et al. concerns a data model inspired by the spiked covariance model of Wang and Fan [47]. They identify a regime of overparameterization where the hard-margin SVM classifier has good generalization (i.e., classification risk going to 0) even when all the training samples are support vectors. Our new lower bound can be regarded as establishing a limit on this approach to the analysis of SVM; specifically, if the (effective) dimension is not sufficiently large, the OLS and SVM solutions may not coincide.

**Prior analyses of number of support vectors.** Besides its relevance to generalization analysis, the number of support vectors in an SVM model is an interesting quantity to study in its own right. Hsu et al. [23] sharpen and extend the analysis of Muthukumar et al. [35] about SVP in the independent features model that we also adopt. They prove that SVM on $n$ samples with $d$ independent subgaussian components coincides with OLS when $d = \Omega(n \log n)$ with probability tending to 1. They also give a converse result stating that the coincidence fails with constant probability when $d = O(n)$ in the isotropic Gaussian feature model. (We give these results here as Theorems 1 and 2 respectively.) Our results generalize and tighten the latter bound to tell an asymptotically sharp story about the phase transition for both isotropic and anisotropic random vectors with subgaussian components. Our specific analysis for the isotropic Gaussian case gives the exact point of the phase transition.

The number of support vectors is also studied in the context of variants of SVM [3, 39], including the soft-margin SVM [12] and the $\nu$-SVM [37]. In these cases, the asymptotic number of support vectors is shown to be related to the noise rate in the problem. The setups we study are linearly separable, which makes it possible to study the hard-margin SVM (without regularization). The hard-margin SVM is also of interest because it captures the implicit bias of gradient descent on the logistic loss objective for linear predictors [25, 38].

Phase transitions have been studied in the context of linear classification [8, 13, 27, 40], and SVMs in particular [7, 14, 28, 30], but most study qualitative changes in behavior other than support vector proliferation. The most relevant is the study of Buhot and Gordon [7], who employ techniques from statistical physics to show the existence of phase transitions for the generalization error, margin size, and number of support vectors as $n$ and $d = \Theta(n)$ become arbitrarily large. While they characterize the fraction of samples that are support vectors, they do not address our question about when *all* samples are support vectors, not just a large fraction. Indeed, our results demonstrate that their regime where $d$ grows linearly with $n$ will not exhibit support vector proliferation when $n$ and $d$ in the limit.

**Overparameterized linear regression.** There has been a recent flurry of analyses of overparameterized linear regression models [e.g., 4, 5, 21, 24, 29, 32–35, 47, 48]. Many of these analyses are carried out in the $d = \Theta(n)$ asymptotic regime, whereas our work studies a phase transition that occurs in a much higher-dimensional regime. The notions of effective dimensions we use are present in the analyses of Bartlett et al. [4] and Muthukumar et al. [35], and the latter work identifies regimes where SVM and OLS coincide and enjoy good performance for both classification and regression.

**High-dimensional geometry and universality.** Our conjecture about support vector proliferation for $\ell_1$-SVMs derives inspiration from studies of high-dimensional geometric phase transitions, particularly those by [1, 2, 15]. These results consider the geometry of random polytopes. Amelunxen and Bürgisser [1] establish phase transitions on the feasibility and boundedness of the solutions to random linear programs, Amelunxen et al. [2] extend these results to characterize when $\ell_1$-norm minimizing solutions to sparse recovery problems are exactly correct, and Donoho and Tanner [15] bound the number of faces of random polytopes. We also borrow heavily from [16] when designing our experiments in Section 5 to test the universality hypothesis.

## 2 Preliminaries

This section introduces notation, as well as the optimization problems and data models we consider. We also define support vector proliferation and prove the equivalence of different formulations.

### 2.1 Notation

For $\lambda \in \mathbb{R}^d_+$, we define the $\ell_2$ and $\ell_\infty$ *dimension proxies* as $d_2 := \|\lambda\|_1^2/\|\lambda\|_2^2$ and $d_\infty := \|\lambda\|_1/\|\lambda\|_\infty$. Let $[n] := \{1, \ldots, n\}$. For some vector $w \in \mathbb{R}^n$ and matrix $A \in \mathbb{R}^{n \times d}$, we let $w_i$ and $A_i$ denote the $i$th element of $w$ and row of $A$ respectively; likewise, we let $A_{.,j} \in \mathbb{R}^n$ represent the $j$th column of $A$. We abuse notation to let $w_{\backslash i} = (w_1, \ldots, w_{i-1}, w_{i+1}, \ldots, w_n) \in \mathbb{R}^{n-1}$, $w_{[m]} = (w_1, \ldots, w_m) \in \mathbb{R}^m$, $w_{\backslash [m]} = (w_{m+1}, \ldots, w_n) \in \mathbb{R}^{n-m}$, and $w_{[m]\backslash i} = (w_{[m]})_{\backslash i} \in \mathbb{R}^{m-1}$ for $i \in [m]$ and $m \in [n]$. Analogous notation holds for $A_{\backslash i}$, $A_{[m]}$, $A_{\backslash [m]}$, and $A_{[m]\backslash i}$. We frequently consider the Gram matrix $\mathbf{K} := \mathbf{X}\mathbf{X}^\intercal \in \mathbb{R}^{n \times n}$ for feature matrix $\mathbf{X} \in \mathbb{R}^{n \times d}$; for these matrices, we let $\mathbf{K}_{\backslash i} = \mathbf{X}_{\backslash i}\mathbf{X}_{\backslash i}^\intercal \in \mathbb{R}^{(n-1) \times (n-1)}$ and analogously define $\mathbf{K}_{[m]}$, $\mathbf{K}_{\backslash [m]}$, and $\mathbf{K}_{[m]\backslash i}$. Let $\mu_{\max}(A)$ and $\mu_{\min}(A)$ represent the largest and smallest eigenvalues of the matrix $A$ respectively, and let $\|A\|$ be the operator norm of $A$. For some vector $y \in \mathbb{R}^n$, we let $\mathrm{diag}(y) \in \mathbb{R}^{n \times n}$ be a diagonal matrix with $(\mathrm{diag}(y))_{i,i} = y_i$. Throughout, boldface characters refer to random variables.

### 2.2 Optimization problems

We consider the hard-margin support vector machine (SVM) optimization problem and ask under what conditions one may expect all the slackness conditions to be satisfied. We consider training samples $(\mathbf{x}_1, y_1), \ldots, (\mathbf{x}_n, y_n) \in \mathbb{R}^d \times \{\pm 1\}$ in a high-dimensional regime where $d \gg n$ with design matrix $\mathbf{X} := [\mathbf{x}_1 | \ldots | \mathbf{x}_n]^\intercal \in \mathbb{R}^{n \times d}$ and Gram matrix $\mathbf{K} = \mathbf{X}\mathbf{X}^\intercal \in \mathbb{R}^{n \times n}$. In full generality, the separating hyperplane corresponding to the $\ell_p$-SVM problem for some $p \geq 1$ is the solution to the following optimization problem:

$$\min_{w \in \mathbb{R}^d} \|w\|_p \quad \text{such that} \quad y_i w^\intercal \mathbf{x}_i \geq 1, \; \forall i \in [n]. \qquad \text{(SVM Primal)}$$

Our results in Sections 3–5 concern $p = 2$, and we discuss $p = 1$ in Section 6. It is worth mentioning that feasibility is not a concern in the settings we consider.[1] An example $(\mathbf{x}_i, y_i)$ is called a *support vector* if it lies exactly on the margin defined by separator $w$, or equivalently if $y_i w^\intercal \mathbf{x}_i = 1$. It is well-known that $w$ can be represented as a non-negative linear combination of all $y_i \mathbf{x}_i$ where $\mathbf{x}_i$ is a support vector [43].

We contrast the weights of the classifier returned by SVM Primal with the weights of minimum $\ell_p$-norm that satisfy the normal equations of ordinary least squares (OLS). In the case where the training data can be linearly interpolated, this optimization problem is:

$$\min_{w \in \mathbb{R}^d} \|w\|_p \quad \text{such that} \quad y_i w^\intercal \mathbf{x}_i = 1, \; \forall i \in [n]. \qquad \text{(Interpolation Primal)}$$

Per the convention mentioned in the introduction, the solution of (Interpolation Primal) when $p = 2$ is referred to as ordinary least squares. Feasibility is ensured as long as the feature vectors $\mathbf{x}_i$ are linearly independent.

### 2.3 Equivalent formulations of SVP

We study the phenomenon of *support vector proliferation* (SVP), i.e., the occurrence in which every example $\mathbf{x}_i$ is a support vector. Because $\mathbf{x}_i$ is a support vector if $y_i w^\intercal \mathbf{x}_i = 1$, this occurs if and only if the solution of (SVM Primal) coincides exactly with that of (Interpolation Primal). Here, we analyze those formulations to show equivalent conditions needed for SVP, which we give in Proposition 1. Before presenting the proposition, we introduce the notation needed to use the alternate formulations.

---

[1]When $d \geq n$, we are always able to find a separating hyperplane since the features are linearly independent with high probability. In fact, a theorem of Cover [13] shows that feasibility holds with high probability under mild distributional assumptions even for $d > n/2$.

We translate the relationship between the two primal optimization problems into the dual space. Taking $\mathbf{A} = \mathrm{diag}(y)\mathbf{X} \in \mathbb{R}^{n \times d}$, the dual of the optimization problem (Interpolation Primal) is:

$$\max_{\alpha \in \mathbb{R}^n} \sum_{i=1}^{n} \alpha_i \quad \text{such that} \quad \|\mathbf{A}^\top \alpha\|_q \leq 1. \qquad \text{(Interpolation Dual)}$$

The dual of (SVM Primal) is (Interpolation Dual) with an additional constraint that $\alpha \in \mathbb{R}^n_+$.

Let $\mathbf{T} = \{\sum_{i=1}^{n} a_i \mathbf{A}_i : \sum_{i=1}^{n} a_i = 1\}$ denote the affine plane passing through the rows of $\mathbf{A}$, and let $\mathbf{T}^+ = \{\sum_{i=1}^{n} a_i \mathbf{A}_i : \sum_{i=1}^{n} a_i = 1, \ a_i \geq 0\}$ be the convex hull of the rows of $\mathbf{A}$. In addition, for $i \in [n]$, let $\mathbf{T}_{\backslash i} = \{\sum_{i' \neq i} a_{i'} \mathbf{A}_{i'} : \sum_{i' \neq i} a_{i'} = 1\}$. We denote $\Pi_{\mathbf{T}}(\mathbf{0})$ as the $\ell_q$-norm projection of the origin onto $\mathbf{T}$, which is uniquely defined for $1 < q < \infty$.

**Proposition 1.** *Let $1 < p < \infty$ and $q = (1 - 1/p)^{-1}$, and consider any $(\mathbf{X}, y) \in \mathbb{R}^{n \times d} \times \{\pm 1\}^n$. Suppose $\mathbf{K}$ is invertible. Then, the following are equivalent:*

*(1) SVP occurs for $\ell_p$-SVM.*

*(2) The solutions $w$ to (SVM Primal) and (Interpolation Primal) are identical.*

*(3) The optimal solution to (Interpolation Dual) lies within the interior of $\mathbb{R}^d_+$.*

*(4) $\Pi_{\mathbf{T}}(\mathbf{0}) \in \mathbf{T}^+$.*

*Moreover, if $p = 2$, then properties (1)–(4) are also equivalent to the following:*

*(5) For all $i \in [n]$, $y_i y_{\backslash i}^\top \mathbf{K}_{\backslash i}^{-1} \mathbf{X}_{\backslash i} \mathbf{x}_i = y_i \Pi_{\mathbf{T}_{\backslash i}}(\mathbf{0})^\top \mathbf{x}_i / \|\Pi_{\mathbf{T}_{\backslash i}}(\mathbf{0})\|_2^2 < 1$.*

This dual framework in (3) and (4) gives an alternative geometric structure to consider for this problem. For the $\ell_2$-case, this formulation draws from the fact that the separating hyperplane obtained from an SVM is represented as a linear combination of support vectors. Although the $\ell_1$ case is not technically covered by Proposition 1 (due to the non-strict convexity of $\ell_1$ norm), our analysis still gives useful insights, and we explore this case specifically in Section 6.

We prove Proposition 1 in Appendix A. The equivalence between (1) and (5) in the $p = 2$ case was proved by Hsu et al. [23, Lemma 1]. Our alternative proof is based on establishing the equivalence of (4) and (5) and draws heavily from our geometric formulation of SVP.

## 2.4 Data model

We use the data model of Hsu et al. [23], where every labeled sample $(\mathbf{x}_i, y_i)$ has $\mathbf{x}_i$ drawn from an anisotropic subgaussian distribution with independent components and arbitrary fixed labels $y_i$.

**Definition.** For some $\lambda \in \mathbb{R}^d_{\geq 0}$, we say $(\mathbf{X}, y) \in \mathbb{R}^{n \times d} \times \{\pm 1\}^n$ (as well as $(\mathbf{X}, \mathbf{Z}, y) \in \mathbb{R}^{n \times d} \times \mathbb{R}^{n \times d} \times \{\pm 1\}^n$) is a $\lambda$-*anisotropic subgaussian sample* if: $y = (y_1, \ldots, y_n) \in \{\pm 1\}^n$ are fixed (non-random) labels; $\mathbf{Z} := [\mathbf{z}_1 | \ldots | \mathbf{z}_n]^\top \in \mathbb{R}^{n \times d}$ is a matrix of independent 1-subgaussian random variables with $\mathbf{E}[\mathbf{z}_{i,j}] = 0$ and $\mathbf{E}[\mathbf{z}_{i,j}^2] = 1$; and $\mathbf{X} := [\mathbf{x}_1 | \ldots | \mathbf{x}_n]^\top = \mathbf{Z} \, \mathrm{diag}(\lambda)^{1/2} \in \mathbb{R}^{n \times d}$. We say $(\mathbf{X}, y)$ is an *isotropic subgaussian sample* if it is $\lambda$-anisotropic for $\lambda = (1, \ldots, 1)$. Finally, we say $(\mathbf{X}, y)$ is an *isotropic Gaussian sample* if it is isotropic subgaussian and each $\mathbf{x}_i \sim \mathcal{N}(0, I_d)$.

We only consider fixed labels $y$ that do not depend on $\mathbf{X}$. However, we do not consider this to be a major limitation of this work. As discussed before, Cover [13] shows that linear separability is overwhelmingly likely in the high-dimensional regimes we consider. Moreover, our results can be extended to a setting where $\mathbf{y}_i = \mathrm{sign}(v^\top \mathbf{x}_i)$ for some fixed weight vector $v$.

## 2.5 Previous results

We tighten and generalize the characterization of the SVP threshold by Hsu et al. [23]. We give versions of their results that are directly comparable to our results in Sections 3 and 4.

**Theorem 1** (Theorem 1 of [23])**.** *Consider a $\lambda$-anisotropic subgaussian sample $(\mathbf{X}, y)$ and any $\delta \in (0, 1)$. If $d_\infty = \Omega(n \log(n) \log(\frac{1}{\delta}))$, then SVP occurs for $\ell_2$-SVM with probability at least $1 - \delta$.*

**Theorem 2** (Theorem 3 of [23])**.** *Consider an isotropic Gaussian sample $(\mathbf{X}, y)$. For some constant $\delta \in (0, 1)$, if $d = O(n)$, then SVP occurs for $\ell_2$-SVM with probability at most $\delta$.*

They note the logarithmic separation between Theorems 1 and 2 and the limitations of the data model used in Theorem 2. The authors pose an improvement in generality and asymptotic tightness to their lower-bound as an open problem, which we resolve in the subsequent sections.

## 3 SVP threshold for anisotropic subgaussian samples

We closely characterize when support vector proliferation does and does not occur through the following theorem, which serves as a converse to Theorem 1.

**Theorem 3** (Lower-bound on SVP threshold for anisotropic subgaussians). *Consider a $\lambda$-anisotropic subgaussian sample $(\mathbf{X}, y)$ and any $\delta \in (0, \frac{1}{2})$. For absolute constants $C_1, C_2, C_3, C_4$, assume that $\lambda$ and $n$ satisfy*

$$n \geq C_1 \left( \log \frac{1}{\delta} \right)^2, \quad d_2 \leq C_2 n \log n, \quad d_\infty \geq C_3 n \log \frac{1}{\delta}, \quad \text{and} \quad d_\infty^2 \geq C_4 d_2 n. \tag{3}$$

*Then, SVP occurs for $\ell_2$-SVM with probability at most $\delta$.*

**Remark 1.** *If each $\mathbf{x}_i$ is drawn from a Gaussian distribution, then we could instead permit $\mathbf{x}_i$ to have any positive semi-definite covariance matrix $\Sigma \in \mathbb{R}^{d \times d}$ with eigenvalues $\lambda_1, \ldots, \lambda_d$ due to rotational invariance.*

**Remark 2.** *In addition, the result can be generalized to subgaussian $\mathbf{x}_i$ with general variance proxies $\gamma \geq 1$. We present the current version for the sake of simplicity and note that the generalization is straightforward.*

In the case where $(\mathbf{X}, y)$ is an isotropic subgaussian sample, Theorem 3 and Theorem 1 (from [23]) together establish that the threshold for SVP occurs at $d = \Theta(n \log n)$. Theorem 3 sharpens and generalizes the partial converse of [23] given in Theorem 2.

Theorem 3 does not depend explicitly on the ambient dimension $d$; instead, it only involves the effective dimension proxies $d_2$ and $d_\infty$, which can be finite even if $d$ is infinite. Thus, the result readily extends to infinite-dimensional Hilbert spaces.

We prove the theorem in Appendix B and briefly summarize the techniques here. By Proposition 1, it suffices to show that with probability $1 - \delta$, $\mathbf{K}$ is invertible and

$$\max_{i \in [n]} y_i y_{\backslash i}^\top \mathbf{K}_{\backslash i}^{-1} \mathbf{X}_{\backslash i} \mathbf{x}_i \geq 1, \tag{4}$$

where $\mathbf{K}_{\backslash i} := \mathbf{X}_{\backslash i} \mathbf{X}_{\backslash i}^\top$. This same equivalence underlies the proof of Theorem 2 in [23]. However, their application of this equivalence is limited because they avoid issues of dependence between random variables by instead lower-bounding the probability that $y_1 y_{\backslash 1}^\top \mathbf{K}_{\backslash 1}^{-1} \mathbf{X}_{\backslash 1} \mathbf{x}_1 \geq 1$. This forces their bound to hold only when $d = O(n)$. We obtain a tighter bound by separating the first $m$ samples (denoted $\mathbf{X}_{[m]}$) for some carefully chosen $m$ and relating the term to the maximum of $m$ independent random variables. To do so, we lower-bound the left-hand side of (4) with the following decomposition:

$$\max_{i \in [m]} \left[ y_i y_{\backslash i}^\top \left( \mathbf{K}_{\backslash i}^{-1} - \frac{1}{\|\lambda\|_1} I_{n-1} \right) \mathbf{X}_{\backslash i} \mathbf{x}_i + \frac{1}{\|\lambda\|_1} y_i y_{[m] \backslash i}^\top \mathbf{X}_{[m] \backslash i} \mathbf{x}_i + \frac{1}{\|\lambda\|_1} y_i y_{\backslash [m]}^\top \mathbf{X}_{\backslash [m]} \mathbf{x}_i \right].$$

We prove that this decomposition (and hence, also (4)) is at least 1 with probability $1 - \delta$ by lower-bounding the three terms with Lemmas 1, 3, and 4 (given in Appendix B). We bound the first two terms for all $i \in [m]$ by employing standard concentration bounds for subgaussian and subexponential random variables and by tightly controlling the spectrum of $\mathbf{K}_{\backslash i}$. To bound the third term, we relate the quantity for each $i \in [m]$ to an independent univariate Gaussian with the Berry-Esseen theorem and apply standard lower-bounds on the maximum of $m$ independent Gaussians.

The assumptions in (3) are all intuitive and necessary for our arguments. The first assumption ensures that enough samples are drawn for high-probability concentration bounds to exist over collections of $n$ variables. The second assumption guarantees the sub-sample size $m$ is sufficiently large to have predictable statistical properties; this is asymptotically tight with its counterpart in Theorem 1 up to a factor of $\log \frac{1}{\delta}$. The third ensures that the variance of each $y_i y_{\backslash i}^\top \mathbf{K}_{\backslash i}^{-1} \mathbf{X}_{\backslash i} \mathbf{x}_i$ term is sufficiently small. The fourth assumption rules out $\lambda$-anisotropic subgaussian distributions with $\|\lambda\|_2^2 \ll \|\lambda\|_\infty^2 n$, where a single component of each $\mathbf{x}_i$ is disproportionately large relative to others and causes unfavorable anti-concentration properties.

## 4   Exact asymptotic threshold for Gaussian samples

Section 3 shows the existence of a change in model behavior when $d = \Theta(n \log n)$ without identifying a precise threshold where this phase transition appears. Here, we refine that analysis for the isotropic Gaussian case to find such an exact threshold. That is, if $d = 2\tau n \log n$, as $n$ becomes large, SVP will occur when $\tau > 1$ and will not occur when $\tau < 1$. Roughly speaking, this phenomenon stems from the fact that terms in (4) are weakly correlated, which causes (4) to behave similarly to a maximum of independent Gaussians. Furthermore, we characterize the rate at which the phase transition sharpens. The following theorem shows that if the convergence $\tau \to 1$ is slow enough, then the asymptotic probabilities of SVP are degenerate and the width of the transition is bounded.

**Theorem 4** (Sharp SVP phase transition). *Let* $(\mathbf{X}, y)$ *be an isotropic Gaussian sample. Let* $(\epsilon_n)_{n \geq 1}$ *be any sequence of positive real numbers such that* $\limsup_{n \to \infty} \epsilon_n < 2 - c_1$ *for some* $c_1 > 0$ *and* $\liminf_{n \to \infty} \epsilon_n \sqrt{\log n} > C_2$ *for some* $C_2 > 0$ *depending only on* $c_1$. *Then,*

$$\lim_{n \to \infty} \mathbf{P}[SVP \text{ occurs for } \ell_2\text{-SVM}] = \begin{cases} 0 & \text{if } d = (2 - \epsilon_n) n \log n, \\ 1 & \text{if } d = (2 + \epsilon_n) n \log n. \end{cases}$$

**Remark 3.** *Theorem 4 characterizes the width of the phase transition: the difference $w_n$ between the values of $d$ where the probability of SVP is (say) 0.9 and 0.1 satisfies $w_n = O(n\sqrt{\log n})$.*

It remains an open problem to determine if this transition width estimate is sharp. Specifically, the bound can be sharpened by exhibiting some sequence $\epsilon_n$ for which the asymptotic probability of support-vector proliferation is non-degenerate.

The proof of Theorem 4 is given in Appendix C. In the case where $d = (2 - \epsilon_n) n \log n$, the proof mirrors that of Theorem 3, but deviates in the final step by using the limiting distribution of the maximum of independent Gaussians. When $d = (2 + \epsilon_n) n \log n$, we follow the basic argument in the proof of Theorem 1 from [23], but we sharpen the analysis by taking advantage of Gaussianity to find the limiting probability as $n \to \infty$.

## 5   Experimental validation of SVP phase transition and universality

While Theorem 4 identifies the exact SVP phase transition for only isotropic Gaussian samples, we demonstrate experimentally that a similarly sharp cutoff occurs for a broader category of data distributions. These experiments suggest that the phase transition phenomenon extends beyond the distributions with independent subgaussian components considered in Theorem 3, and that it occurs at the same location ($d = 2n \log n$), with the transition sharpening as $n \to \infty$.

Our simulation procedure is as follows. We generate data sets $(\mathbf{X}, y) \in \mathbb{R}^{n \times d} \times \{\pm 1\}^n$, where $y \in \{\pm 1\}^n$ is a fixed vector of labels with exactly $n/2$ positive labels, and $\mathbf{x}_1, \ldots, \mathbf{x}_n \sim_{\text{i.i.d.}} \mathcal{D}^{\otimes d}$ where $\mathcal{D}$ is one of six sample distributions on $\mathbb{R}$. For each data set, we check for SVP by solving the problem in (Interpolation Primal), and checking if its solution additionally satisfies the constraints from (SVM Primal). Let $\hat{\mathbf{p}} := \hat{\mathbf{p}}(n, d; \mathcal{D}, M)$ denote the observed frequency of SVP in $M = 400$ independent trials when $\mathbf{X} \in \mathbb{R}^{n \times d}$ is generated using $\mathcal{D}$. (Full details are given in Appendix D.1.)

Figure 1 shows a heat map of $\hat{\mathbf{p}}(n, d; \mathcal{D}, M)$ with $M = 400$. The striking similarity across the distributions suggests that SVP is a universal phenomenon for a broad class of sample distributions that vary qualitatively in different aspects: biased vs. unbiased, continuous vs. discrete, bounded vs. unbounded, and subgaussian vs. non-subgaussian. Moreover, the boundary at which the sharp transition occurs is visibly indistinguishable across the different sample distributions.

We also investigate the universality of SVP using statistical methodology inspired by Donoho and Tanner [16]. Specifically, we model $\hat{\mathbf{p}}$ using Probit regression to test our *universality hypothesis*: that the occurrence of SVP for $\ell_2$-SVM on data with independent features matches the behavior under Gaussian features as $n$ and $d$ grow large. Our model is $M \cdot \hat{\mathbf{p}} \sim \text{Binom}(p(n, d; \mathcal{D}), M)$, where

$$p(n, d; \mathcal{D}) = \Phi\left(\mu^{(0)}(n, \mathcal{D}) + \mu^{(1)}(n, \mathcal{D}) \times \tau + \mu^{(2)}(n, \mathcal{D}) \times \log \tau\right)$$

$$\text{with} \quad \mu^{(i)}(n, \mathcal{D}) = \mu_0^{(i)}(\mathcal{D}) + \frac{\mu_1^{(i)}(\mathcal{D})}{\sqrt{n}}.$$

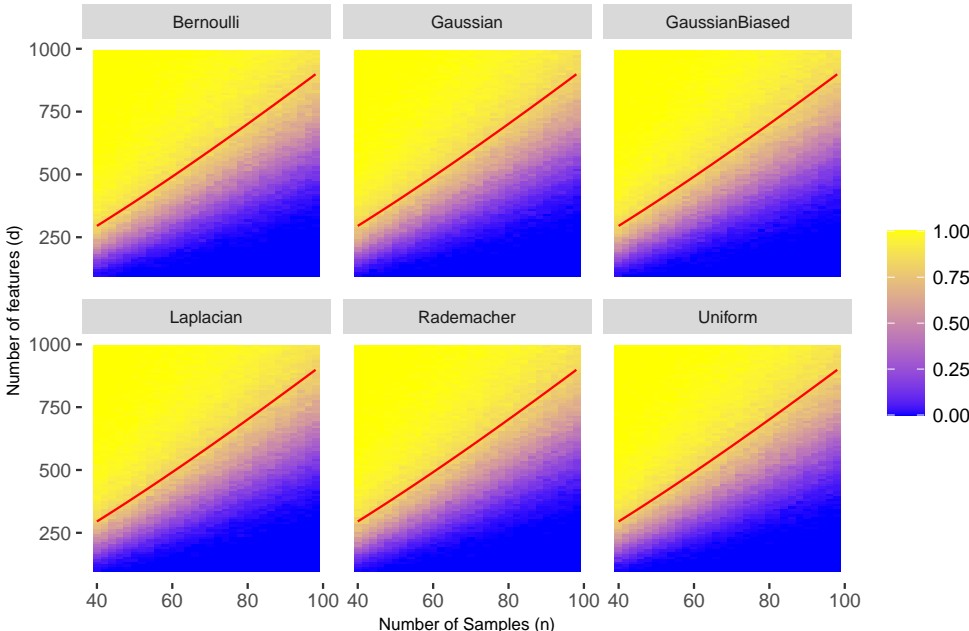

Figure 1: The fraction of $M = 400$ trials where support vector proliferation occurs for $n$ samples, $d$ features, and six different sample distributions $\mathcal{D}$. All distributions demonstrate sharp phase transitions near the theoretical boundary $n \mapsto 2n \log n$, illustrated by the red curve.

Here, $\Phi(t)$ is the standard normal distribution function, $\tau = d/(2n \log n)$, and the model parameters are $\mu_0^{(i)}(\mathcal{D})$ and $\mu_1^{(i)}(\mathcal{D})$ for $i \in \{0, 1, 2\}$ and the six different distributions $\mathcal{D}$ (shown in Table 1 in Appendix D.1). Figure 2 visualizes the fitted Probit function $p$ for fixed $n$ and $\tau$ and demonstrates that the model provides a very accurate approximation of $\hat{\mathbf{p}}$.

The universality hypothesis corresponds to the model in which the parameters $\mu_0^{(i)}(\mathcal{D})$ are "tied together" (i.e., forced to be the same) for all distributions $\mathcal{D}$. That is, only the parameters scaled down by a factor of $\sqrt{n}$, $\mu_1^{(i)}(\mathcal{D})$, are allowed to vary with $\mathcal{D}$. The scaling ensures that their effect tends to zero as $n \to \infty$. The alternative (non-universality) hypothesis corresponds to the model in which all parameters (both $\mu_0^{(i)}(n, \mathcal{D})$ and $\mu_1^{(i)}(n, \mathcal{D})$ for each $i$) are allowed to vary with $\mathcal{D}$. We compare the models' goodness-of-fit using analysis of deviance [22]. Our main finding is that the experimental data are consistent with the universality hypothesis (and also that we can reject a null hypothesis in which all parameters are "tied together" for all $\mathcal{D}$). The details and model diagnostics are given in Appendix D.2.

Finally, in Appendix D.3, we provide empirical support for the generality of Remark 3, namely that the transition width is roughly $n\sqrt{\log n}$ for data models other than Gaussian ensembles.

## 6  SVP phase transition for $\ell_1$-SVMs?

Because both the SVM and linear regression problems can be formulated for general $\ell_p$-norms, we can ask similar questions about when their solutions coincide. Here, we examine the $\ell_1$ case: the coincidence of SVM with an $\ell_1$-penalty and $\ell_1$-norm minimizing interpolation (also called Basis Pursuit [11]). Linear models with $\ell_1$ regularization are often motivated by the desire for sparse weight vectors [e.g., 11, 34, 36, 41].

Based on experimental evidence and the differences in high-dimensional geometry between $\ell_\infty$ and $\ell_2$ balls, we conjecture that SVP for $\ell_1$-SVMs only occurs in a much higher-dimensional regime.

**Conjecture 1.** *Let $(\mathbf{X}, y) \in \mathbb{R}^{n \times d} \times \mathbb{R}^n$ be an isotropic Gaussian sample. Then, the probability of SVP occurring for an $\ell_1$-SVM with $\mathbf{X}$ and $y$ undergoes a phase transition around $d = f(n)$, for some*

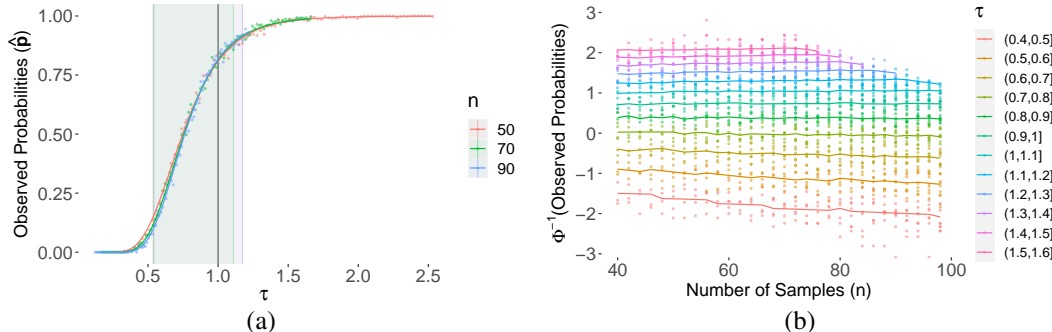

(a)                                                                (b)

Figure 2: Visualizations of SVP frequencies for constant slices of $n$ and $\tau$ for $d = 2\tau n \log n$. *Left panel (a):* The points are $(\tau, \hat{\mathbf{p}})$ from the Gaussian samples, for fixing $n \in \{50, 70, 90\}$. The black vertical line corresponds to $\tau = 1$. The Probit model's predictions are overlaid, and shaded regions correspond to $\tau$ for which the model's predicted probabilities are between $0.1$ and $0.9$. *Right panel (b):* The points show $(n, \Phi^{-1}(\hat{\mathbf{p}}))$ from a Gaussian distribution, fixing $\tau$ to lie in one of 12 different intervals. The Probit model's predictions (averaged over all $\tau$ within an interval) are overlaid.

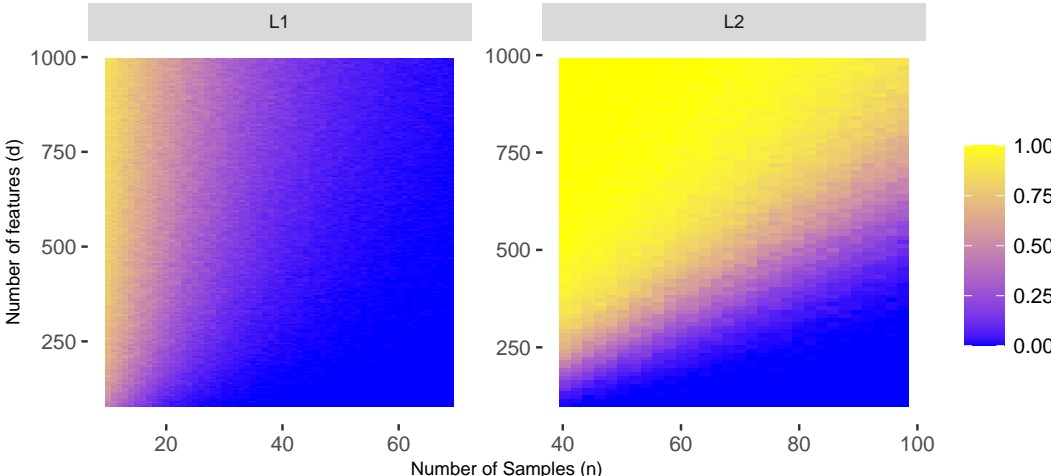

Figure 3: The observed probabilities of support vector proliferation for $\ell_1$- and $\ell_2$-SVMs for $d$-dimensional isotropic Gaussian samples of size $n$.

$f(n) = \omega(n \log n)$. *Formally, there exist positive constants $c$ and $c'$ with $c \leq c'$ such that*

$$\lim_{n \to \infty} \mathbf{P}\left[\text{SVP occurs for } \ell_1\text{-SVM}\right] = \begin{cases} 0 & \text{if } d < cf(n), \\ 1 & \text{if } d > c'f(n). \end{cases}$$

Conjecture 1 is consistent with our preliminary experimental findings, summarized in Figure 3. It shows larger values of $d$ relative to $n$ are needed to ensure SVP for $\ell_1$-SVMs and that the transition appears to be less sharp. Indeed, the experiments indicate that the true phase transition may occur when $d$ is asymptotically *much* larger than $n \log n$. They do not rule out the possibility that the transition may even require $d = \exp(\Omega(n))$. Further experimental details are given in Appendix E.1.

Answering whether support vector proliferation occurs in the $\ell_1$ case is equivalent to determining whether the optimal solution $\alpha^*$ to (Interpolation Dual) problem lies in the positive orthant $\mathbb{R}_+^n$.[2] In the $\ell_1$ case, we have $q = \infty$, so solving the problem amounts to characterizing the solutions to the

---

[2]While Proposition 1 does not imply this equivalence for $\ell_1$-SVMs for arbitrary data, the results of the proposition are valid for isotropic Gaussian samples, because the corresponding $\ell_\infty$ projection $\Pi_{\mathbf{T}}(\mathbf{0})$ in those cases is well-defined almost surely.

following linear program for data matrix $\mathbf{A} = \mathrm{diag}(y)\mathbf{X} \in \mathbb{R}^{n \times d}$:

$$\max_{\alpha \in \mathbb{R}^n} \quad \sum_{i=1}^{n} \alpha_i \quad \text{s.t.} \quad -\mathbf{1} \le \mathbf{A}^{\mathsf{T}}\alpha \le \mathbf{1}. \tag{Dual L1}$$

There is a line of work that gives high probability guarantees about whether related random linear programs are feasible and where their solutions reside [1, 2]. Similar analyses of random linear programs may be useful for understanding how large $d$ must be to have $\alpha^* \in \mathbb{R}_+^n$, and we carry out a preliminary characterization in Appendix E.

Conjecture 1 and the other questions raised in this work point to a broader scope of investigations about high-dimensional phenomena and universality concerning optimization problems commonly used in machine learning and statistics. Our results, along with those from prior works, provide new analytic and empirical approaches that may prove useful in tackling these questions.

## Acknowledgments and Disclosure of Funding

D. Hsu acknowledges support from NSF grants CCF-1740833 and IIS-1563785, NASA ATP grant 80NSSC18K109, and a Sloan Research Fellowship. C. Sanford acknowledges support from NSF grant CCF-1563155 and a Google Faculty Research Award to D. Hsu. N. Ardeshir acknowledges support from Columbia Statistics Department. This material is based upon work supported by the National Science Foundation under grant numbers listed above. Any opinions, findings and conclusions or recommendations expressed in this material are those of the authors and do not necessarily reflect the views of the National Science Foundation. We acknowledge computing resources from Columbia University's Shared Research Computing Facility project, which is supported by NIH Research Facility Improvement Grant 1G20RR030893-01, and associated funds from the New York State Empire State Development, Division of Science Technology and Innovation (NYSTAR) Contract C090171, both awarded April 15, 2010.

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
