# Support vector machines and linear regression coincide with very high-dimensional features

**Navid Ardeshir\***
Dept. of Statistics
Columbia University
na2844@columbia.edu

**Clayton Sanford\***
Dept. of Computer Science
Columbia University
clayton@cs.columbia.edu

**Daniel Hsu**
Dept. of Computer Science
Columbia University
djhsu@cs.columbia.edu

## Abstract

The support vector machine (SVM) and minimum Euclidean norm least squares regression are two fundamentally different approaches to fitting linear models, but they have recently been connected in models for very high-dimensional data through a phenomenon of support vector proliferation, where every training example used to fit an SVM becomes a support vector. In this paper, we explore the generality of this phenomenon and make the following contributions. First, we prove a super-linear lower bound on the dimension (in terms of sample size) required for support vector proliferation in independent feature models, matching the upper bounds from previous works. We further identify a sharp phase transition in Gaussian feature models, bound the width of this transition, and give experimental support for its universality. Finally, we hypothesize that this phase transition occurs only in much higher-dimensional settings in the $\ell_1$ variant of the SVM, and we present a new geometric characterization of the problem that may elucidate this phenomenon for the general $\ell_p$ case.

## 1  Introduction

The *support vector machine* (SVM) and *ordinary least squares* (OLS) are well-weathered approaches to fitting linear models, but they are associated with different learning tasks: classification and regression. In this paper, we study the case in which the models return exactly the same hypothesis for sufficiently high-dimensional data.

The hard-margin SVM is a linear classification model that finds the separating hyperplane that maximizes the minimum margin of error for every training sample. If the training data $(\mathbf{x}_1, y_1), \ldots, (\mathbf{x}_n, y_n) \in \mathbb{R}^d \times \{\pm 1\}$ are linearly separable, then the resulting linear classifier is $x \mapsto \text{sign}\,(x^\mathsf{T} w_{\text{SVM}})$, where $w_{\text{SVM}}$ is the solution to the following optimization problem:

$$w_{\text{SVM}} = \underset{w \in \mathbb{R}^d}{\arg\min} \|w\|_2 \quad \text{such that} \quad y_i w^\mathsf{T} \mathbf{x}_i \geq 1, \ \forall i \in [n]. \tag{1}$$

An example $\mathbf{x}_i$ is a *support vector* if the corresponding constraint is satisfied with equality, and the optimal solution $w_{\text{SVM}}$ is a linear combination of these support vectors.

Ordinary least squares regression finds the linear function that best fits the training data $(\mathbf{x}_1, y_1), \ldots, (\mathbf{x}_n, y_n) \in \mathbb{R}^d \times \mathbb{R}$ according to the sum of squared errors. When the solution is not unique, it is natural to take the solution of minimum Euclidean norm; this is the convention we adopt. Taking $\mathbf{X} := [\mathbf{x}_1 | \ldots | \mathbf{x}_n]^\mathsf{T} \in \mathbb{R}^{n \times d}$ and $y := (y_1, \ldots, y_n)$, the solution is the hypothesis $x \mapsto w_{\text{OLS}}^\mathsf{T} x$ where $w_{\text{OLS}}$ is the solution to the following: $w_{\text{OLS}} = \arg\min_{w \in \mathbb{R}^d} \|w\|_2$ such that $\mathbf{X}^\mathsf{T}\mathbf{X}w = \mathbf{X}^\mathsf{T}y$. In many high-dimensional settings (e.g., where $\mathbf{X}$ has full row rank), the solution may in fact *interpolate* the training data, i.e.,

$$w_{\text{OLS}} = \underset{w \in \mathbb{R}^d}{\arg\min} \|w\|_2 \quad \text{such that} \quad w^\mathsf{T} \

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

[ \textit{SVP occurs for } \ell_1\textit{-SVM} \right] = \begin{cases} 0 & \textit{if } d < cf(n), \\ 1 & \textit{if } d > c'f(n). \end{cases}$$

Conjecture 1 is consistent with our preliminary experimental findings, summarized in Figure 3. It shows larger values of $d$ relative to $n$ are needed to ensure SVP for $\ell_1$-SVMs and that the transition appears to be less sharp. Indeed, the experiments indicate that the true phase transition may occur when $d$ is asymptotically *much* larger than $n \log n$. They do not rule out the possibility that the transition may even require $d = \exp(\Omega(n))$. Further experimental details are given in Appendix E.1.

Answering whether support vector proliferation occurs in the $\ell_1$ case is equivalent to determining whether the optimal solution $\alpha^*$ to (Interpolation Dual) problem lies in the positive orthant $\mathbb{R}^n_+$.[2] In the $\ell_1$ case, we have $q = \infty$, so solving the problem amounts to characterizing the solutions to the

---

[2]While Proposition 1 does not imply this equivalence for $\ell_1$-SVMs for arbitrary data, the results of the proposition are valid for isotropic Gaussian samples, because the corresponding $\ell_\infty$ projection $\Pi_{\mathbf{T}}(\mathbf{0})$ in those cases is well-defined almost surely.

following linear program for data matrix $\mathbf{A} = \mathrm{diag}(y)\mathbf{X} \in \mathbb{R}^{n \times d}$:

$$\max_{\alpha \in \mathbb{R}^n} \quad \sum_{i=1}^n \alpha_i \quad \text{s.t.} \quad -\mathbf{1} \le \mathbf{A}^\top \alpha \le \mathbf{1}. \qquad \text{(Dual L1)}$$

There is a line of work that gives high probability guarantees about whether related random linear programs are feasible and where their solutions reside [1, 2]. Similar analyses of random linear programs may be useful for understanding how large $d$ must be to have $\alpha^* \in \mathbb{R}_+^n$, and we carry out a preliminary characterization in Appendix E.

Conjecture 1 and the other questions raised in this work point to a broader scope of investigations about high-dimensional phenomena and universality concerning optimization problems commonly used in machine learning and statistics. Our results, along with those from prior works, provide new analytic and empirical approaches that may prove useful in tackling these questions.

## Acknowledgments and Disclosure of Funding

D. Hsu acknowledges support from NSF grants CCF-1740833 and IIS-1563785, NASA ATP grant 80NSSC18K109, and a Sloan Research Fellowship. C. Sanford acknowledges support from NSF grant CCF-1563155 and a Google Faculty Research Award to D. Hsu. N. Ardeshir acknowledges support from Columbia Statistics Department. This material is based upon work supported by the National Science Foundation under grant numbers listed above. Any opinions, findings and conclusions or recommendations expressed in this material are those of the authors and do not necessarily reflect the views of the National Science Foundation. We acknowledge computing resources from Columbia University's Shared Research Computing Facility project, which is supported by NIH Research Facility Improvement Grant 1G20RR030893-01, and associated funds from the New York State Empire State Development, Division of Science Technology and Innovation (NYSTAR) Contract C090171, both awarded April 15, 2010.

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

# A  Proofs for Section 2

We restate and prove Proposition 1.

**Proposition 1.** *Let $1 < p < \infty$ and $q = (1 - 1/p)^{-1}$, and consider any $(\mathbf{X}, y) \in \mathbb{R}^{n \times d} \times \{\pm 1\}^n$. Suppose $\mathbf{K}$ is invertible. Then, the following are equivalent:*

*(1) SVP occurs for $\ell_p$-SVM.*
*(2) The solutions $w$ to (SVM Primal) and (Interpolation Primal) are identical.*
*(3) The optimal solution to (Interpolation Dual) lies within the interior of $\mathbb{R}_+^d$.*
*(4) $\Pi_{\mathbf{T}}(\mathbf{0}) \in \mathbf{T}^+$.*

*Moreover, if $p = 2$, then properties (1)–(4) are also equivalent to the following:*

*(5) For all $i \in [n]$, $y_i y_{\backslash i}^{\mathsf{T}} \mathbf{K}_{\backslash i}^{-1} \mathbf{X}_{\backslash i} \mathbf{x}_i = y_i \Pi_{\mathbf{T}_{\backslash i}}(\mathbf{0})^{\mathsf{T}} \mathbf{x}_i / \|\Pi_{\mathbf{T}_{\backslash i}}(\mathbf{0})\|_2^2 < 1$.*

*Proof.* The proof henceforth proceeds under the assumption that $\mathbf{K}$ is invertible. Since $\mathbf{K}$ is symmetric, the invertibility of $\mathbf{K}$ implies that all of its principal minors (i.e., the $\mathbf{K}_{\backslash i}$'s) are invertible.

The equivalences between the first four statements follow from simple implications of the definition of support vector proliferation and the derivation of the dual optimization problems to (SVM Primal) and (Interpolation Primal). Lemma 1 of [23] proves the equivalence between (1) and (5) for the $\ell_2$ case and completes the argument. We supplement the argument with an additional equivalence between (4) and (5) to show how the "leave-one-out terms" in (5) can be intuitively understood through the geometric framing of the dual problem.

**(1) $\iff$ (2):**  This is immediate from the fact that the definition of SVP corresponds exactly to the equality constraints that are present in (Interpolation Primal) and not in (SVM Primal).

**(2) $\iff$ (3):**  This equivalence follows by deriving the duals of the two optimization problems, (Interpolation Primal) and (SVM Primal), and by noting that the only difference between the corresponding duals is that the latter has an additional requirement that $\alpha \in \mathbb{R}_+^d$.

By adding Lagrange multipliers $\alpha \in \mathbb{R}^n$, we obtain the dual of (Interpolation Primal):

$$\max_{\alpha \in \mathbb{R}^n} \sum_{i=1}^n \alpha_i + \min_{w \in \mathbb{R}^d} \|w\|_p - w^{\mathsf{T}} \sum_{i=1}^n \alpha_i y_i \mathbf{x}_i.$$

By Holder's inequality, if we take $q$ to be the dual of $p$ (i.e. $\frac{1}{p} + \frac{1}{q} = 1$), then $|w^{\mathsf{T}} u| \leq \|w\|_p \|u\|_q$, with equality when $|u_i|^q$ is proportional to $|w_i|^p$. Therefore,

$$\min_{w \in \mathbb{R}^d} \|w\|_p - w^{\mathsf{T}} u = \begin{cases} 0 & \|u\|_q \leq 1 \\ -\infty & \text{otherwise.} \end{cases}$$

We further denote $\mathbf{A} = \mathrm{diag}(y)\mathbf{X} \in \mathbb{R}^{n \times d}$, whose $i$th row is $y_i \mathbf{x}_i$. We conclude that the dual of the optimization problem (Interpolation Primal) is exactly (Interpolation Dual). We similarly find that the dual of (SVM Primal) is

$$\max_{\alpha \in \mathbb{R}_+^n} \sum_{i=1}^n \alpha_i \quad \text{such that} \quad \|\mathbf{A}^{\mathsf{T}} \alpha\|_q \leq 1. \tag{SVM Dual}$$

Because (Interpolation Dual) and (SVM Dual) coincide if and only if the $\alpha$ that solves the former is in the positive orthant ($\alpha \in \mathbb{R}_+^d$), the equivalence follows.

**(3) $\iff$ (4):**  We reconfigure (Interpolation Dual) to rewrite the optimization problem as a projection. Let $\Pi_{\mathbf{T}}$ and $\Pi_{\mathbf{T}^+}$ be the $\ell_q$-norm minimizing projection operators onto $\mathbf{T}$ and $\mathbf{T}^+$, which are

uniquely defined when $\mathbf{K}$ is invertible. Then the solution to (Interpolation Dual) is:[3]

$$\min_{\alpha \in \mathbb{R}^n} \left\| \mathbf{A}^\mathsf{T} \frac{\alpha}{\sum_{i=1}^n \alpha_i} \right\|_q = \|\Pi_\mathbf{T}(\mathbf{0})\|_q. \qquad \text{(Interpolation Projection)}$$

By the definition of $\mathbf{T}$ and $\mathbf{T}^+$ and the fact that $\Pi_\mathbf{T}(\mathbf{0}) = \mathbf{A}^\mathsf{T}\alpha^*/\mathbf{1}^\mathsf{T}\alpha^*$ for $\alpha^*$ optimizing (Interpolation Projection), we have that $\alpha^* \in \mathbb{R}_+^d$ if and only if $\Pi_\mathbf{T}(\mathbf{0}) \in \mathbf{T}^+$.

**(4)** $\iff$ **(5):** By the (Interpolation Projection) formulation, $\Pi_\mathbf{T}(\mathbf{0})$ can be alternatively interpreted as the projection of the origin onto the affine space $\mathbf{T}$. Therefore we have,

$$\Pi_\mathbf{T}(\mathbf{0}) = \sum_{i=1}^n a_i^* \mathbf{A}_i = \arg\min_{u \in \mathbf{T}} \|u\|_2^2 \qquad (5)$$

where $a_i^* \in \mathbb{R}$ is proportional to $\alpha_i^*$ such that $\sum_{i=1}^n a_i^* = 1$. This is possible because the optimal value of (Interpolation Primal) is positive.

The following steps show the equivalence:

1. For every $i \in [n]$ we show,

$$a_i^* > 0 \quad \iff \quad \mathbf{A}_i^\mathsf{T} \Pi_{\mathbf{T}_{\backslash i}}(\mathbf{0}) < \left\| \Pi_{\mathbf{T}_{\backslash i}}(\mathbf{0}) \right\|_2^2 \qquad (6)$$

   by leveraging the fact that $\ell_2$ space is equipped with inner product which allows us to decompose the contribution of each sample to the projection.

2. We find an explicit expression for $\Pi_\mathbf{T}(\mathbf{0})$:

$$\Pi_\mathbf{T}(\mathbf{0}) = \frac{\mathbf{A}^\mathsf{T} (\mathbf{A}\mathbf{A}^\mathsf{T})^{-1} \mathbf{1}}{\mathbf{1}^\mathsf{T} (\mathbf{A}\mathbf{A}^\mathsf{T})^{-1} \mathbf{1}} = \frac{\mathbf{X}(\mathbf{X}^\mathsf{T}\mathbf{X})^{-1} y}{y^\mathsf{T}(\mathbf{X}^\mathsf{T}\mathbf{X})^{-1} y}. \qquad (7)$$

   An analogous expression for $\Pi_{\mathbf{T}_{\backslash i}}(\mathbf{0})$ can be found for any fixed $i$ by the same method, which gives the desired equivalence by combining with (6).

*First Step:* Fix some index $i \in [n]$. Since $\Pi_{\mathbf{T}_{\backslash i}}(\mathbf{A}_i)$ is on the affine space $\mathbf{T}_{\backslash i}$, which is closed under affine linear combination, one can express $\mathbf{T}$ in the following way:

$$\mathbf{T} = \left\{ \sum_{j=1}^n a_j \mathbf{A}_j : \sum_{j=1}^n a_j = 1 \right\}$$
$$= \left\{ a_i \left( \mathbf{A}_i - \Pi_{\mathbf{T}_{\backslash i}}(\mathbf{A}_i) \right) + \left( a_i \Pi_{\mathbf{T}_{\backslash i}}(\mathbf{A}_i) + \sum_{j \neq i} a_j \mathbf{A}_j \right) : \sum_{j \neq i} a_j = 1 - a_i\,, a_i \in \mathbb{R} \right\}$$
$$= \left\{ a_i (\mathbf{A}_i - \Pi_{\mathbf{T}_{\backslash i}}(\mathbf{A}_i)) + u : u \in \mathbf{T}_{\backslash i}\,, a_i \in \mathbb{R} \right\}.$$

By the definition of the projection onto $\mathbf{T}$, we represent $\Pi_\mathbf{T}(\mathbf{0}) = a^* \left( \mathbf{A}_i - \Pi_{\mathbf{T}_{\backslash i}}(\mathbf{A}_i) \right) + u^*$ where,

$$(a^*, u^*) = \arg\min_{(a,u) \in \mathbb{R} \times \mathbf{T}_{\backslash i}} \left\| a(\mathbf{A}_i - \Pi_{\mathbf{T}_{\backslash i}}(\mathbf{A}_i)) + u \right\|_2^2.$$

It is straightforward to see that $a^* = a_i^*$ by comparing this representation with equation (5) alongside with the fact that $\mathbf{A}_i - \Pi_{\mathbf{T}_{\backslash i}}(\mathbf{A}_i)$ is orthogonal to $\mathbf{T}_{\backslash i}$:

$$\mathbf{0} = \Pi_\mathbf{T}(\mathbf{0}) - \Pi_\mathbf{T}(\mathbf{0}) = (a^* - a_i^*) \left( \mathbf{A}_i - \Pi_{\mathbf{T}_{\backslash i}}(\mathbf{A}_i) \right) + \left( u^* - \underbrace{\left( \sum_{j \neq i} a_j^* \mathbf{A}_j + a_i^* \Pi_{\mathbf{T}_{\backslash i}}(\mathbf{0}) \right)}_{\in \mathbf{T}_{\backslash i}} \right).$$

---

[3]Note that (Interpolation Dual) will never be optimized by $\mathbf{0}$, because there must always exist some $\alpha$ with strictly positive components such that $\|\mathbf{A}^\mathsf{T}\alpha\|_q \leq 1$. Therefore, we need not worry about the projection being undefined.

To find $a^*$, we sequentially optimize over $u$, substitute its optimal value, and then optimize over $a$. It suffices to minimize $\|u\|_2^2$ because $u^\intercal(\mathbf{A}_i - \Pi_{\mathbf{T}_{\backslash i}}(\mathbf{A}_i))$ is constant for all $u \in \mathbf{T}_{\backslash i}$ due to orthogonality and the definition of $\mathbf{T}_{\backslash i}$. Hence, $u^* = \Pi_{\mathbf{T}_{\backslash i}}(\mathbf{0})$ is optimal. Subsequently, by optimizing over $a$ and setting the derivative to zero,

$$(\mathbf{A}_i - \Pi_{\mathbf{T}_{/i}}(\mathbf{A}_i))^\intercal \left( a^* \left( \mathbf{A}_i - \Pi_{\mathbf{T}_{/i}}(\mathbf{A}_i) \right) + \Pi_{\mathbf{T}_{\backslash i}}(\mathbf{0}) \right) = 0$$

or equivalently,

$$a^* = \frac{(\Pi_{\mathbf{T}_{/i}}(\mathbf{A}_i) - \mathbf{A}_i)^\intercal \Pi_{\mathbf{T}_{\backslash i}}(\mathbf{0})}{\left\| \mathbf{A}_i - \Pi_{\mathbf{T}_{\backslash i}}(\mathbf{A}_i) \right\|_2^2} = \frac{\left\| \Pi_{\mathbf{T}_{\backslash i}}(\mathbf{0}) \right\|_2^2 - \mathbf{A}_i^\intercal \Pi_{\mathbf{T}_{\backslash i}}(\mathbf{0})}{\left\| \mathbf{A}_i - \Pi_{\mathbf{T}_{/i}}(\mathbf{A}_i) \right\|_2^2}$$

For the last step, we combine the facts that $\Pi_{\mathbf{T}_{\backslash i}}(\mathbf{0})^\intercal u$ is constant over $\mathbf{T}_{\backslash i}$ and $\Pi_{\mathbf{T}_{\backslash i}}(\mathbf{0}) \in \mathbf{T}_{\backslash i}$. Note that the denominator is non-zero because the invertibility of $\mathbf{K}$ implies that $\mathbf{A}_i$ will not lie on the span of the remaining rows $\mathbf{A}_{\backslash i}$, and hence will not be in $\mathbf{T}_{\backslash i}$. This immediately proves (6).

*Second Step:* We now shift our focus to expressing $\Pi_{\mathbf{T}}(\mathbf{0})$ explicitly in terms of $\mathbf{A}$. As discussed in Section 2, the projection of the origin onto $\mathbf{T}$ for $\ell_2$-SVM can be expressed in an explicit way as $\Pi_{\mathbf{T}}(\mathbf{0}) = \mathbf{A}^\intercal \alpha^* / \mathbf{1}^\intercal \alpha^*$, where $\alpha^*$ is the unique solution to (Interpolation Dual) for $p = q = 2$. In order to represent $\alpha^*$ explicitly one can reform (Interpolation Dual) using a simple change of variables $\nu = (\mathbf{A}\mathbf{A}^\intercal)^{1/2} \alpha$ into,

$$\nu^* = \arg\max_{\nu \in \mathbb{R}^n} \nu^\intercal (\mathbf{A}\mathbf{A}^\intercal)^{-1/2} \mathbf{1} \quad \text{s.t. } \|\nu\|_2 \le 1.$$

This problem can be easily understood using a simple Cauchy-Schwarz inequality,

$$\alpha^* = (\mathbf{A}\mathbf{A}^\intercal)^{-1/2} \nu^* = (\mathbf{A}\mathbf{A}^\intercal)^{-1} \mathbf{1} / \sqrt{\mathbf{1}^\intercal (\mathbf{A}\mathbf{A}^\intercal)^{-1} \mathbf{1}}.$$

In conclusion, we can express the projection as,

$$\Pi_{\mathbf{T}}(\mathbf{0}) = \frac{\mathbf{A}^\intercal \alpha^*}{\mathbf{1}^\intercal \alpha^*} = \frac{\mathbf{A}^\intercal (\mathbf{A}\mathbf{A}^\intercal)^{-1} \mathbf{1}}{\mathbf{1}^\intercal (\mathbf{A}\mathbf{A}^\intercal)^{-1} \mathbf{1}} = \frac{\mathbf{X}^\intercal (\mathbf{X}\mathbf{X}^\intercal)^{-1} y}{y^\intercal (\mathbf{X}\mathbf{X}^\intercal)^{-1} y}. \qquad \square$$

## B   Proofs for Section 3

We prove Theorem 3, which we restate below.

**Theorem 3** (Lower-bound on SVP threshold for anisotropic subgaussians). *Consider a $\lambda$-anisotropic subgaussian sample $(\mathbf{X}, y)$ and any $\delta \in (0, \frac{1}{2})$. For absolute constants $C_1, C_2, C_3, C_4$, assume that $\lambda$ and $n$ satisfy*

$$n \ge C_1 \left( \log \frac{1}{\delta} \right)^2, \quad d_2 \le C_2 n \log n, \quad d_\infty \ge C_3 n \log \frac{1}{\delta}, \quad and \quad d_\infty^2 \ge C_4 d_2 n. \tag{3}$$

*Then, SVP occurs for $\ell_2$-SVM with probability at most $\delta$.*

*Proof.* As discussed in Section 3, it suffices to prove that $\mathbf{K}_{\backslash i}$ is invertible for all $i$ and

$$\max_{i \in [m]} \left[ y_i y_{\backslash i}^\intercal \left( \mathbf{K}_{\backslash i}^{-1} - \frac{1}{\|\lambda\|_1} I_{n-1} \right) \mathbf{X}_{\backslash i} \mathbf{x}_i + \frac{1}{\|\lambda\|_1} y_i y_{[m]\backslash i}^\intercal \mathbf{X}_{[m]\backslash i} \mathbf{x}_i + \frac{1}{\|\lambda\|_1} y_i y_{\backslash[m]}^\intercal \mathbf{X}_{\backslash[m]} \mathbf{x}_i \right] \ge 1$$

with probability $1 - \delta$ for some $m \le n$. We do so by showing that the following three events each hold with probability $1 - \frac{\delta}{3}$:

$$\max_{i \in [m]} \left| y_i y_{\backslash i}^\intercal \left( \mathbf{K}_{\backslash i}^{-1} - \frac{1}{\|\lambda\|_1} I_{n-1} \right) \mathbf{X}_{\backslash i} \mathbf{x}_i \right| \le 1,$$

$$\max_{i \in [m]} \frac{1}{\|\lambda\|_1} \left| y_i y_{[m]\backslash i}^\intercal \mathbf{X}_{[m]\backslash i} \mathbf{x}_i \right| \le 1, \text{ and}$$

$$\max_{i \in [m]} \frac{1}{\|\lambda\|_1} y_i y_{\backslash[m]}^\intercal \mathbf{X}_{\backslash[m]} \mathbf{x}_i \ge 3.$$

It remains to plug in the results of Lemmas 1, 3, and 4 to show that the three events occur with high probability given the conditions imposed on $n, d_2, d_\infty$, and $\delta$ in (3). Let $m := \lceil \exp(\frac{d_2}{2C_2 n}) \rceil$. By (3), $m \le \sqrt{n} + 1 \le \frac{n}{\log n} \le \frac{n}{2}$ for sufficiently small constant $C_1$.

1. By Lemma 1 in Appendix B.1 with $\delta := \frac{\delta}{3m}$, it follows that for any fixed $i \in [m]$, $\mathbf{K}_{\backslash i}$ is invertible and

$$\left| y_i y_{\backslash i}^\mathsf{T} \left( \mathbf{K}_{\backslash i}^{-1} - \frac{1}{\|\lambda\|_1} I_{n-1} \right) \mathbf{X}_{\backslash i} \mathbf{x}_i \right| \leq 1$$

with probability at least $1 - \frac{\delta}{3m}$ as long as the following conditions hold:

$$d_\infty \geq c_3' \left( n \left( \log \frac{3m}{\delta} \right)^{1/3} + n^{1/3} \log \frac{3m}{\delta} \right), \tag{8}$$

$$d_2 d_\infty \geq c_4 n \log \frac{3m}{\delta} \left( n + \log \frac{3m}{\delta} \right). \tag{9}$$

We show that the inequalities in (8) and (9) are implied by the preconditions of Theorem 3 in (3) by choosing sufficiently large constants $C_1$, $C_3$, and $C_4$. For (8),

$$n \left( \log \frac{3m}{\delta} \right)^{1/3} + n^{1/3} \log \frac{3m}{\delta} \leq n \left( \frac{d_2}{C_2 n} + \log \frac{3}{\delta} \right)^{1/3} + n^{1/3} \log \frac{3n}{2\delta}$$

$$\leq \frac{n^{2/3} d_2^{1/3}}{C_2} + n \left( \log \frac{3}{\delta} \right)^{1/3} + n \log \frac{3}{\delta}$$

$$\leq \frac{n^{1/3} d_\infty^{2/3}}{C_2 C_4^{1/3}} + 2n \log \frac{3}{\delta},$$

which implies the desired inequality. To establish (9):

$$n \log \frac{3m}{\delta} \left( n + \log \frac{3m}{\delta} \right) \leq \frac{d_2}{C_2} \left( n + \log \frac{3n}{2\delta} \right) \leq \frac{d_2}{C_2} \left( 2n + \log \frac{1}{\delta} \right) \leq \frac{d_2}{C_2} \cdot \frac{3 d_\infty}{C_3}.$$

By applying a union bound to all $m$ events, they all occur with probability at least $1 - \frac{\delta}{3}$.

2. By applying Lemma 3 in Appendix B.2 for all $i \in [m]$ with $\delta$ as before and union-bounding over the corresponding events, with probability $1 - \frac{\delta}{3}$,

$$\frac{1}{\|\lambda\|_1} y_i y_{[m]\backslash i}^\mathsf{T} \mathbf{X}_{[m]\backslash i} \mathbf{x}_i \geq -1$$

for all $i \in [m]$ as long as

$$d_2 \geq c_1 m \log \frac{3m}{\delta} \quad \text{and} \quad d_\infty \geq c_2 \sqrt{m} \log \frac{3m}{\delta}.$$

Both inequalities follow from the third inequality of (3) for sufficiently large $C_3$ and by $m \leq \frac{n}{\log n}$.

3. By Lemma 4 in Appendix B.3 with $t := 3$,

$$\max_{i \in [m]} \frac{1}{\|\lambda\|_1} y_i y_{\backslash [m]}^\mathsf{T} \mathbf{X}_{\backslash [m]} \mathbf{x}_i \geq 3$$

with probability $1 - \frac{\delta}{3}$, if

$$n - m \geq c_1 \left( \log \frac{3}{\delta} \right)^2, \qquad \exp \left( \frac{9 d_2}{c_2 (n-m)} \right) \leq m \leq (n-m),$$

$$d_2 \geq c_3 (n-m) \log \log \frac{3}{\delta}, \qquad \text{and} \quad d_\infty^2 \geq c_4 d_2 (n-m).$$

The inequalities are satisfied as immediate consequences of (3) and the fact that $m = \lceil \exp(\frac{d_2}{2 C_2 n}) \rceil \leq \frac{n}{2}$ for sufficiently small $C_2$. $\qquad \square$

In the subsequent three sections, we prove Lemmas 1, 3, and 4.

## B.1 Bounded difference between leave-one-out terms with $\mathbf{K}^{-1}$ and scaled identity

**Lemma 1.** *Let $(\mathbf{X}, y) \in \mathbb{R}^{n \times d} \times \mathbb{R}^n$ and $(\mathbf{x}', y') \in \mathbb{R}^d \times \mathbb{R}$ be $\lambda$-anisotropic subgaussian samples that are independent of one another with $\mathbf{K} = \mathbf{X}\mathbf{X}^\intercal$. Pick any $\delta \in (0, \frac{1}{2})$. There exist universal constants $c_3, c$ such that if $d_\infty \geq c_3(n + \log \frac{1}{\delta})$, then with probability $1 - \delta$, $\mathbf{K}$ is invertible and*

$$\left| y^\intercal \left( \mathbf{K}^{-1} - \frac{1}{\|\lambda\|_1} I_n \right) \mathbf{X}\mathbf{x}' \right| \leq c \sqrt{\frac{n \log \frac{1}{\delta}}{d_\infty}} \left( \sqrt{\frac{n + \log \frac{1}{\delta}}{d_2}} + \frac{n + \log \frac{1}{\delta}}{d_\infty} \right).$$

**Remark 4.** *To guarantee that $|y^\intercal (\mathbf{K}^{-1} - \|\lambda\|_1^{-1} I_n)\mathbf{X}\mathbf{x}'| \leq \epsilon$ with probability $1 - \delta$ for some $\epsilon > 0$, it suffices to show that*

$$d_\infty \geq c_3' \cdot \frac{n \left( \log \frac{1}{\delta} \right)^{1/3} + n^{1/3} \log \frac{1}{\delta}}{\epsilon^{2/3}} \quad and \quad d_2 d_\infty \geq c_4 \cdot \frac{n \log \frac{1}{\delta} \left( n + \log \frac{1}{\delta} \right)}{\epsilon^2},$$

*for universal constants $c_3'$ and $c_4$.*

The proof of Lemma 1 relies heavily on a concentration bound on the eigenvalues of the Gram matrix $\mathbf{K}$, which draws from a technical lemma of Hsu et al. [23]. We present and prove this result below and then use it to prove Lemma 1.

**Lemma 2.** *Let $(\mathbf{X}, y) \in \mathbb{R}^{n \times d} \times \mathbb{R}$ be $\lambda$-anisotropic subgaussian samples with Gram matrix $\mathbf{K} := \mathbf{X}\mathbf{X}^\intercal \in \mathbb{R}^{n \times n}$. Pick any $\delta \in (0, \frac{1}{2})$. For some universal constant $c$, with probability $1 - \delta$,*

$$\|\mathbf{K} - \|\lambda\|_1 I_n\| \leq c \|\lambda\|_1 \left( \sqrt{\frac{n + \log \frac{1}{\delta}}{d_2}} + \frac{n + \log \frac{1}{\delta}}{d_\infty} \right), \tag{10}$$

*where $\|\cdot\|$ denotes the spectral (operator) norm. If additionally $d_\infty \geq c_3(n + \log \frac{1}{\delta})$ for some universal constant $c_3$, then for the same event, $\mathbf{K}$ is invertible and*

$$\left\| \mathbf{K}^{-1} - \frac{1}{\|\lambda\|_1} I_n \right\| \leq \frac{c}{\|\lambda\|_1} \left( \sqrt{\frac{n + \log \frac{1}{\delta}}{d_2}} + \frac{n + \log \frac{1}{\delta}}{d_\infty} \right). \tag{11}$$

*Proof.* Equation (10) follows from Lemma 8 of Hsu et al. [23]. For some universal constant $c'$ and sufficiently large $c$, we have the following:

$$\mathbf{P} \left[ \|\mathbf{K} - \|\lambda\|_1 I_n\| \geq c \|\lambda\|_1 \left( \sqrt{\frac{n + \log \frac{1}{\delta}}{d_2}} + \frac{n + \log \frac{1}{\delta}}{d_\infty} \right) \right]$$

$$\leq 2 \cdot 9^n \cdot \exp \left( -c' \min \left( \frac{c^2 \|\lambda\|_1^2 (n + \log \frac{1}{\delta})}{\|\lambda\|_2^2 \, d_2}, \frac{c \|\lambda\|_1 (n + \log \frac{1}{\delta})}{\|\lambda\|_\infty \, d_\infty} \right) \right)$$

$$\leq 2 \exp \left( n \log 9 - c' \min(c^2, c) \left( n + \log \frac{1}{\delta} \right) \right) \leq \delta.$$

If equation (11) holds and $c_3$ is sufficiently large, then all eigenvalues of $\mathbf{K}$ are strictly positive and $\mathbf{K}$ is invertible. We now derive equation (11) by bounding the eigenvalues of $\mathbf{K}^{-1}$, assuming that the

event in equation (10) occurs, and rescaling $c$:

$$\left\| \mathbf{K}^{-1} - \frac{1}{\|\lambda\|_1} I_n \right\|$$

$$\leq \max \left( \mu_{\max}(\mathbf{K}^{-1}) - \frac{1}{\|\lambda\|_1}, -\mu_{\min}(\mathbf{K}^{-1}) + \frac{1}{\|\lambda\|_1} \right)$$

$$= \max \left( \frac{1}{\mu_{\min}(\mathbf{K})} - \frac{1}{\|\lambda\|_1}, -\frac{1}{\mu_{\max}(\mathbf{K})} + \frac{1}{\|\lambda\|_1} \right)$$

$$\leq \frac{1}{\|\lambda\|_1} \max \left( \frac{1}{1 - c\left( \sqrt{\frac{n+\log\frac{1}{\delta}}{d_2}} + \frac{n+\log\frac{1}{\delta}}{d_\infty} \right)} - 1, 1 - \frac{1}{1 + c\left( \sqrt{\frac{n+\log\frac{1}{\delta}}{d_2}} + \frac{n+\log\frac{1}{\delta}}{d_\infty} \right)} \right)$$

$$\leq \frac{1}{\|\lambda\|_1} \cdot 2c \left( \sqrt{\frac{n+\log\frac{1}{\delta}}{d_2}} + \frac{n+\log\frac{1}{\delta}}{d_\infty} \right). \qquad \square$$

*Proof of Lemma 1.* Conditioned on $\mathbf{X}$, $y^\mathsf{T}(\mathbf{K}^{-1} - \|\lambda\|_1^{-1} I_n)\mathbf{X}\mathbf{x}'$ is a univariate subgaussian random variable with mean 0 and variance proxy at most $\|y^\mathsf{T}(\mathbf{K}^{-1} - \|\lambda\|_1^{-1} I_n)\mathbf{X}\|^2 \|\lambda\|_\infty$, as long as $\mathbf{K}$ is invertible. We bound the variance proxy and show that $\mathbf{K}$ is invertible with high probability by applying Lemma 2 for some universal $c'$ with probability $1 - \frac{\delta}{2}$:

$$\left\| y^\mathsf{T}(\mathbf{K}^{-1} - \frac{1}{\|\lambda\|_1} I_n)\mathbf{X} \right\|^2 \|\lambda\|_\infty \leq \|y\|_2^2 \left\| \mathbf{K}^{-1} - \frac{1}{\|\lambda\|_1} I_n \right\|^2 \|\mathbf{K}\| \|\lambda\|_\infty$$

$$\leq n \cdot \frac{c'}{\|\lambda\|_1^2} \left( \sqrt{\frac{n+\log\frac{1}{\delta}}{d_2}} + \frac{n+\log\frac{1}{\delta}}{d_\infty} \right)^2 \cdot 2\|\lambda\|_1 \|\lambda\|_\infty$$

$$= \frac{2c'n}{d_\infty} \left( \sqrt{\frac{n+\log\frac{1}{\delta}}{d_2}} + \frac{n+\log\frac{1}{\delta}}{d_\infty} \right)^2$$

We observe that the bound holds for a proper choice of $c$ for a standard concentration bound for a subgaussian random variable. $\qquad \square$

### B.2 Concentration of leave-one-out terms

**Lemma 3.** *Let $(\mathbf{X}, \mathbf{Z}, y) \in \mathbb{R}^{n\times d} \times \mathbb{R}^{n\times d} \times \mathbb{R}^n$ and $(\mathbf{x}', \mathbf{z}', y') \in \mathbb{R}^d \times \mathbb{R}^d \times \mathbb{R}$ be $\lambda$-anisotropic subgaussian samples that are independent of one another. Pick any $\delta \in (0, \frac{1}{2})$. There exists a universal constant $c$ such that with probability $1 - \delta$,*

$$\frac{1}{\|\lambda\|_1} |y^\mathsf{T}\mathbf{X}\mathbf{x}'| \leq c \left( \sqrt{\frac{n\log\frac{1}{\delta}}{d_2}} + \frac{\sqrt{n}\log\frac{1}{\delta}}{d_\infty} \right).$$

**Remark 5.** *To ensure that $\|\lambda\|_1^{-1} |y^\mathsf{T}\mathbf{X}\mathbf{x}'| \leq \epsilon$ with probability $1 - \delta$ for some $\epsilon > 0$, it suffices to show that*

$$d_2 \geq \frac{c_1 n\log\frac{1}{\delta}}{\epsilon^2} \quad \text{and} \quad d_\infty \geq \frac{c_2\sqrt{n}\log\frac{1}{\delta}}{\epsilon}$$

*for universal constants $c_1, c_2$.*

*Proof.* We can rewrite $y^\mathsf{T}\mathbf{X}\mathbf{x}'$ for some vector of 1-subgaussian random variables $\tilde{\mathbf{z}} \in \mathbb{R}^d$.

$$y^\mathsf{T}\mathbf{X}\mathbf{x}' = y^\mathsf{T}\mathbf{Z}\mathrm{diag}(\lambda)\mathbf{z}' = \sqrt{n}\tilde{\mathbf{z}}^\mathsf{T}\mathrm{diag}(\lambda)\mathbf{z}' = \sqrt{n}\sum_{i=1}^{d} \lambda_i \tilde{\mathbf{z}}_i \mathbf{z}'_i.$$

By Lemma 2.7.7 of [45], each $\tilde{\mathbf{z}}_i \mathbf{z}_i'$ is an independent $(1, 2)$-subexponential random variable. Thus, $\sum_{i=1}^d \lambda_i \tilde{\mathbf{z}}_i \mathbf{z}_i'$ is $(\|\lambda\|_2^2, 2\|\lambda\|_\infty)$-subexponential, and with probability $1 - \delta$,

$$|y^\mathsf{T} \mathbf{X} \mathbf{x}'| \leq \sqrt{n}c \left( \sqrt{\|\lambda\|_2^2 \log \frac{1}{\delta}} + \|\lambda\|_\infty \log \frac{1}{\delta} \right).$$

We have the claim by dividing by $\|\lambda\|_1$. $\qquad\qquad\square$

### B.3 Anti-concentration for independent leave-one-out terms

**Lemma 4.** *Let* $(\mathbf{X}, \mathbf{Z}, y) \in \mathbb{R}^{n \times d} \times \mathbb{R}^{n \times d} \times \mathbb{R}^n$ *and* $(\mathbf{X}', \mathbf{Z}', y') \in \mathbb{R}^{m \times d} \times \mathbb{R}^{m \times d} \times \mathbb{R}^m$ *be* $\lambda$-*anisotropic subgaussian samples that are independent of each other. Pick any* $t > 0$ *and* $\delta \in (0, \frac{1}{2})$. *For universal constants* $c_1$, $c_2$, $c_3$, *and* $c_4$, *if*

$$n \geq c_1 \left( \log \frac{1}{\delta} \right)^2, \quad \exp\left( \frac{t^2 d_2}{c_2 n} \right) \leq m \leq n, \quad d_2 \geq c_3 n \log \log \frac{1}{\delta}, \quad \textit{and} \quad d_\infty^2 \geq c_4 d_2 n,$$

*then with probability* $1 - \delta$

$$\max_{i \in [m]} \frac{1}{\|\lambda\|_1} y^\mathsf{T} \mathbf{X} \mathbf{x}_i' \geq t.$$

*Proof.* Let $\mathbf{q}_j := \|\lambda\|_1^{-1} y^\mathsf{T} \mathbf{Z}_{\cdot, j}$ for $j \in [d]$, where $\mathbf{Z}_{\cdot, j} = (\mathbf{z}_{1,j}, \dots, \mathbf{z}_{n,j}) \in \mathbb{R}^n$. Note that

$$\frac{1}{\|\lambda\|_1} y^\mathsf{T} \mathbf{X} \mathbf{x}_i' = \sum_{j=1}^d \mathbf{q}_j \lambda_j \mathbf{z}_{i,j}'$$

and all $\mathbf{q}_j$ and $\mathbf{z}_{i,j}'$ are independent and that $\mathbf{q}_j$ is subgaussian with variance proxy $n/\|\lambda\|_1^2$. The proof of the claim is powered by the Berry-Esseen theorem and comes in two parts:

- We first show that $\mathbf{X}$ is well-behaved with high probability; we say that $\mathbf{X}$ is *good* if the following hold:

$$\sum_{j=1}^d \lambda_j^2 \mathbf{q}_j^2 \geq \frac{n}{2d_2}, \tag{12}$$

$$\max_j |\lambda_j \mathbf{q}_j| \leq \frac{n^{1/4}}{\sqrt{d_2}}. \tag{13}$$

We prove that $\mathbf{P}[\mathbf{X} \text{ is good}] \geq 1 - \frac{2\delta}{3}$.
- Then, we use the Berry-Esseen Theorem to show that, for each $i \in [m]$,

$$\mathbf{P}\left[ \frac{1}{\|\lambda\|_1} y^\mathsf{T} \mathbf{X} \mathbf{x}_i' \leq t \mid \mathbf{X} \text{ is good} \right] \leq \left( \frac{\delta}{3} \right)^{1/m}.$$

Because $\|\lambda\|_1^{-1} y_i' y^\mathsf{T} \mathbf{X} \mathbf{x}_i'$ for $i \in [m]$ are conditionally independent given $\mathbf{X}$, we obtain the desired statement:

$$\mathbf{P}\left[ \max_{i \in [m]} \frac{1}{\|\lambda\|_1} y_i' y^\mathsf{T} \mathbf{X} \mathbf{x}_i' \leq t \right] \leq \mathbf{P}\left[ \mathbf{X} \text{ is not good} \right] + \mathbf{P}\left[ \max_{i \in [m]} \frac{1}{\|\lambda\|_1} y_i' y^\mathsf{T} \mathbf{X} \mathbf{x}_i' \leq t \mid \mathbf{X} \text{ is good} \right]$$

$$\leq \frac{2\delta}{3} + \prod_{i=1}^m \mathbf{P}\left[ \frac{1}{\|\lambda\|_1} y_i' y^\mathsf{T} \mathbf{X} \mathbf{x}_i' \leq t \mid \mathbf{X} \text{ is good} \right] \leq \delta.$$

**X satisfies** (12) **with high probability.** The claim follows from the Hanson-Wright inequality [45], the lower-bound on $n$ with respect to $\delta$, the assumption $d_\infty^2 \geq c_4 d_2 n$, and the fact $\|\lambda\|_4^4 \leq \|\lambda\|_2^2 \|\lambda\|_\infty^2$.

$$\mathbf{P}\left[\sum_{j=1}^{d} \lambda_j^2 \mathbf{q}_j^2 \leq \frac{n}{2d_2}\right] \leq \mathbf{P}\left[\left|\sum_{j=1}^{d} \lambda_j^2 \mathbf{q}_j^2 - \mathbf{E}\left[\sum_{j=1}^{d} \lambda_j^2 \mathbf{q}_j^2\right]\right| \leq \frac{n}{2d_2}\right]$$

$$\leq 2\exp\left(-c'' \min\left(\frac{n^2}{4d_2^2} \cdot \frac{\|\lambda\|_1^4}{n^2} \cdot \frac{1}{\|\lambda\|_4^4}, \frac{n}{2d_2} \cdot \frac{\|\lambda\|_1^2}{n} \cdot \frac{1}{\|\lambda\|_\infty^2}\right)\right)$$

$$\leq 2\exp\left(-\frac{c'' d_\infty^2}{4d_2}\right) \leq 2\exp\left(-\frac{c'' c_4 n}{4}\right) \leq \frac{\delta}{3},$$

for sufficiently large absolute constant $c_4$.

**X satisfies** (13) **with high probability.** We introduce a different sequence of random variables $\mathbf{r}_1, \ldots, \mathbf{r}_n$ to eliminate any dependence on $d$. Set $0 = k_0 < k_1 < \cdots < k_n = d$ such that for all $i \in [n]$,

$$\sum_{j=k_{i-1}+1}^{k_i} \lambda_j^2 \leq \frac{2\|\lambda\|_2^2}{n}.$$

Such a partition is possible because we can require that $\|\lambda\|_\infty^2 \leq \|\lambda\|_2^2/n$ by assuming $c_4$ is large enough. Let

$$\mathbf{r}_i = \sum_{j=k_{i-1}+1}^{k_i} \lambda_j^2 \mathbf{q}_j^2.$$

Note that all $\mathbf{r}_i$ are independent and that

$$\sum_{j=1}^{d} \lambda_j^2 \mathbf{q}_j^2 = \sum_{i=1}^{n} \mathbf{r}_i.$$

Because $\mathbf{E}[\mathbf{q}_j] = 0$ and $\mathbf{E}[\mathbf{q}_j^2] = n/\|\lambda\|_1^2$, we can bound $\mathbf{E}[\mathbf{r}_i]$:

$$\mathbf{E}[\mathbf{r}_i] \leq \frac{2n\|\lambda\|_2^2}{n\|\lambda\|_1^2} \leq \frac{2}{d_2}.$$

We upper-bound $\max_j |\lambda_j \mathbf{q}_j|$ by noting that $\max_j |\lambda_j \mathbf{q}_j| \leq \max_i \sqrt{\mathbf{r}_i}$. By the Hanson-Wright inequality, the subgaussianity of $\mathbf{q}_i$, and the lower-bound on $n$ with respect to $\delta$,

$$\mathbf{P}\left[\mathbf{r}_i \geq \frac{\sqrt{n}}{d_2}\right] \leq \mathbf{P}\left[\mathbf{r}_i \geq \mathbf{E}[\mathbf{r}_i] + \frac{\sqrt{n}}{2d_2}\right]$$

$$\leq 2\exp\left(-c' \min\left(\frac{n}{4d_2^2} \cdot \frac{\|\lambda\|_1^4}{n^2} \cdot \frac{1}{\sum_{j=k_{i-1}+1}^{k_i} \lambda_j^4}, \frac{\sqrt{n}}{2d_2} \frac{\|\lambda\|_1^2}{n} \frac{1}{\max_{k_{i-1}<j\leq k_i} \lambda_j^2}\right)\right)$$

$$\leq 2\exp\left(-c' \min\left(\frac{n}{4}, \frac{\sqrt{n}}{2}\right)\right) \leq \frac{\delta}{3n}$$

for some absolute constants $c'$ and for a sufficiently large setting of $c_1$. Thus, (13) is satisfied with probability $1 - \frac{\delta}{3}$ by a union bound.

**Bound on term given good X.** We use the Berry-Esseen theorem [6] to relate the maximization over $\|\lambda\|_1^{-1} y^\intercal \mathbf{X} \mathbf{x}_1'$ to a maximization over standard Gaussians. Consider some fixed good $\mathbf{X}$ (and hence, fixed $\mathbf{q}_j$ for all $j \in [d]$). Then, for some absolute constant $c$ and for univariate standard Gaussian $\mathbf{g}$,

$$\sup_{t\in\mathbb{R}}\left|\mathbf{P}\left[\frac{\|\lambda\|_1^{-1} y^\intercal \mathbf{X} \mathbf{x}_1'}{\sqrt{\sum_{j=1}^{d} \mathbf{q}_j^2 \lambda_j^2 \mathbf{E}[\mathbf{z}_{i,j}'^2]}} \leq t\right] - \mathbf{P}[\mathbf{g} \leq t]\right| \leq \frac{c}{\sqrt{\sum_{j=1}^{d} \mathbf{q}_j^2 \lambda_j^2 \mathbf{E}[\mathbf{z}_{i,j}'^2]}} \cdot \max_{j\in[d]} \frac{|\mathbf{q}_j^3| \lambda_j^3 \mathbf{E}[|\mathbf{z}_{i,j}'^3|]}{\mathbf{q}_j^2 \lambda_j^2 \mathbf{E}[\mathbf{z}_{i,j}'^2]}.$$

Because each $\mathbf{z}_{i,j}$ is subgaussian, $\mathbf{E}[\mathbf{z}_{i,j}^3] \le \rho = O(1)$ for some $\rho$. We simplify the expression by plugging in the second and third moments of $\mathbf{z}'_{i,j}$ and rescaling $t$:

$$\sup_{t \in \mathbb{R}} \left| \mathbf{P}\left[ \frac{1}{\|\lambda\|_1} y^\intercal \mathbf{X} \mathbf{x}'_1 \le t \right] - \mathbf{P}\left[ \mathbf{g} \le \frac{t}{\sqrt{\sum_{j=1}^d \lambda_j^2 \mathbf{q}_j^2}} \right] \right| \le \frac{c\rho}{\sqrt{\sum_{j=1}^d \lambda_j^2 \mathbf{q}_j^2}} \cdot \max_{j \in [d]} |\mathbf{q}_j \lambda_j|.$$

Because we assume that $\mathbf{X}$ is good, we plug in our upper-bound on $\max_j |\lambda_j \mathbf{q}_j|$ and lower-bound on $\sum_{j=1}^d \lambda_j^2 \mathbf{q}_j^2$.

$$\mathbf{P}\left[ \frac{1}{\|\lambda\|_1} y^\intercal \mathbf{X} \mathbf{x}'_1 \le t \right] \le \mathbf{P}\left[ \mathbf{g} \le \frac{t}{\sqrt{\sum_{j=1}^d \lambda_j^2 \mathbf{q}_j^2}} \right] + \frac{c\rho}{\sqrt{\sum_{j=1}^d \lambda_j^2 \mathbf{q}_j^2}} \cdot \max_{j \in [d]} |\lambda_j \mathbf{q}_j|$$

$$\le \mathbf{P}\left[ \mathbf{g} \le \frac{t\sqrt{2d_2}}{\sqrt{n}} \right] + \frac{c\rho\sqrt{2d_2}}{\sqrt{n}} \cdot \frac{n^{1/4}}{\sqrt{d_2}}$$

$$\le \mathbf{P}\left[ \mathbf{g} \le \frac{t\sqrt{2d_2}}{\sqrt{n}} \right] + \frac{\sqrt{2}c\rho}{n^{1/4}}$$

We now bound the first term by invoking the Mills ratio bound for the Gaussian distribution function (Fact 1) with the assumption that $c_2 \le \frac{1}{8}$. The remainder of the inequalities follow by enforcing that $c_1$ and $c_2$ be sufficiently large and small respectively:

$$\mathbf{P}\left[ \mathbf{g} \le \frac{t\sqrt{2d_2}}{\sqrt{n}} \right] \le \mathbf{P}\left[ \mathbf{g} \le \sqrt{\frac{1}{4}\log m} \right]$$

$$\le 1 - \frac{1}{\sqrt{2\pi}}\left( \frac{1}{\sqrt{\log(m)/4}} - \frac{1}{(\log(m)/4)^{3/2}} \right) \exp\left( -\frac{\log m}{8} \right)$$

$$\le 1 - \frac{1}{m^{1/6}} \le 1 - \frac{1}{\sqrt{m}} - \frac{\sqrt{2}c\rho}{n^{1/4}}.$$

Therefore,

$$\mathbf{P}\left[ \frac{1}{\|\lambda\|_1} y^\intercal \mathbf{X} \mathbf{x}'_1 \le t \mid \mathbf{X} \text{ is good} \right] \le 1 - \frac{1}{\sqrt{m}} \le \exp\left( -\frac{1}{\sqrt{m}} \right) \le \left( \frac{\delta}{3} \right)^{1/m},$$

which above holds when $m \ge (\log \frac{3}{\delta})^2$ and is ensured by a sufficiently large choice of $c_3$. $\qquad\square$

The following well-known fact is the Mills ratio bound.

**Fact 1.** *Let $\Phi$ denote the standard Gaussian distribution function. Then for any $t \ge 0$,*

$$\left( \frac{1}{t} - \frac{1}{t^3} \right) \cdot \frac{1}{\sqrt{2\pi}} e^{-t^2/2} \le 1 - \Phi(t) \le \frac{1}{t} \cdot \frac{1}{\sqrt{2\pi}} e^{-t^2/2}.$$

### B.4 Modification of Theorem 3 to support dependent labels

While an apparent weakness of Theorem 3 is the fixed labels $y$, this can be surmounted. Here, we outline how the proof can be easily modified to include labels $\mathbf{y}_i = \text{sign}(v^\intercal \mathbf{x}_i)$ for some unit vector $v$ for the isotropic Gaussian case; we believe this can be further generalized, but we present this version for the sake of simplicity.

**Theorem 5** (Lower-bound on SVP threshold with dependent labels). *Fix any $v \in \mathbb{R}^d$ with $\|v\| = 1$, and consider an isotropic Gaussian sample $(\mathbf{X}, \mathbf{y})$ where each $\mathbf{y}_i = v^\intercal \mathbf{x}_i$ and any $\delta \in (0, \frac{1}{2})$ For absolute constants $C_1, C_2, C_3, C_4$, assume that $d$ and $n$ satisfy*

$$n \ge C_1 \left( \log \frac{1}{\delta} \right)^2, \quad d \le C_2 n \log n, \quad \text{and} \quad d \ge C_3 n \log \frac{1}{\delta}, \tag{14}$$

*Then[4], SVP occurs for $\ell_2$-SVM with probability at most $\delta$.*

---

[4]The fourth constraint is omitted because $d_2 = d_\infty = d$ in the isotropic case.

The proof of the theorem relies on the same decomposition as Theorem 3. The first step (Lemma 1) proceeds identically, because the proof of the lemma uses no other properties of the $\mathbf{y}$ besides the fact that it belongs to $\{\pm 1\}^n$. The following inequality allows the remainder of the proof to proceed identically by fixing $y = \mathbf{1}$. For all $t \in \mathbb{R}$ and $i \in [n]$,

$$\mathbf{P}\left[y_i \mathbf{y}_{\backslash i}^\mathsf{T} \mathbf{X}_{\backslash i} \mathbf{x}_i \geq t\right] \geq \mathbf{P}\left[\mathbf{1}^\mathsf{T} \mathbf{X}_{\backslash i} \mathbf{x}_i \geq t\right].$$

We can prove this fact for the simple Gaussian setting by taking advantage of the fact that orthogonal components of a spherical Gaussian are independent. For all $i$, we write $\mathbf{x}_i = (v^\mathsf{T} \mathbf{x}_i)v + \mathbf{x}_i'$ where $v^\mathsf{T} \mathbf{x}_i' = 0$. Then

$$\mathbf{y}_i \mathbf{y}_{\backslash i}^\mathsf{T} \mathbf{X}_{\backslash i} \mathbf{x}_i = \sum_{j \neq i} (\mathbf{y}_j \mathbf{x}_j)^\mathsf{T} (\mathbf{y}_i \mathbf{x}_i) = \sum_{j \neq i} \operatorname{sign}\left(v^\mathsf{T} \mathbf{x}_j v^\mathsf{T} \mathbf{x}_i\right) \left[v^\mathsf{T} \mathbf{x}_j v^\mathsf{T} \mathbf{x}_i \|v\|_2^2 + \mathbf{x}_j'^\mathsf{T} \mathbf{x}_i'\right]$$

$$= \sum_{j \neq i} [|v^\mathsf{T} \mathbf{x}_j v^\mathsf{T} \mathbf{x}_i| + \operatorname{sign}\left(v^\mathsf{T} \mathbf{x}_j v^\mathsf{T} \mathbf{x}_i\right) \mathbf{x}_j'^\mathsf{T} \mathbf{x}_i'] \geq \sum_{j \neq i} [v^\mathsf{T} \mathbf{x}_j v^\mathsf{T} \mathbf{x}_i + \operatorname{sign}\left(v^\mathsf{T} \mathbf{x}_j v^\mathsf{T} \mathbf{x}_i\right) \mathbf{x}_j'^\mathsf{T} \mathbf{x}_i'].$$

By independence and symmetry, each term in the last sum is distributed identically to $v^\mathsf{T} \mathbf{x}_j v^\mathsf{T} \mathbf{x}_i + \mathbf{x}_j'^\mathsf{T} \mathbf{x}_i' = \mathbf{x}_j^\mathsf{T} \mathbf{x}_i$. This gives the claim.

## C  Proofs for Section 4

In this section, we give the proof of Theorem 4.

**Theorem 4** (Sharp SVP phase transition). *Let* $(\mathbf{X}, y)$ *be an isotropic Gaussian sample. Let* $(\epsilon_n)_{n \geq 1}$ *be any sequence of positive real numbers such that* $\limsup_{n \to \infty} \epsilon_n < 2 - c_1$ *for some* $c_1 > 0$ *and* $\liminf_{n \to \infty} \epsilon_n \sqrt{\log n} > C_2$ *for some* $C_2 > 0$ *depending only on* $c_1$. *Then,*

$$\lim_{n \to \infty} \mathbf{P}[\textit{SVP occurs for } \ell_2\textit{-SVM}] = \begin{cases} 0 & \textit{if } d = (2 - \epsilon_n) n \log n, \\ 1 & \textit{if } d = (2 + \epsilon_n) n \log n. \end{cases}$$

We divide the proof into two cases, which we each prove in the two following subsections.

### C.1  Below the threshold

We first consider the case where the dimension is below the threshold, specifically $d = (2 - \epsilon_n) n \log n$.

Our proof follows the same strategy as that of Theorem 3. Let $m := n / \log n$ and assume $n$ is sufficiently large so that $m \leq n/2$. Using the equivalence from Proposition 1, it suffices to show that the following event has probability tending to 1:

$$\max_{i \in [m]} \left[y_i \mathbf{y}_{\backslash i}^\mathsf{T} \left(\mathbf{K}_{\backslash i}^{-1} - \frac{1}{d} I_{n-1}\right) \mathbf{X}_{\backslash i} \mathbf{x}_i + \frac{1}{d} y_i y_{[m] \backslash i}^\mathsf{T} \mathbf{X}_{[m] \backslash i} \mathbf{x}_i + \frac{1}{d} y_i \mathbf{y}_{\backslash [m]}^\mathsf{T} \mathbf{X}_{\backslash [m]} \mathbf{x}_i\right] \geq 1.$$

**Lemma 5.** *For any* $C_0 > 0$ *and* $c_1 \in (0, 2)$, *there exists* $C_2 > 0$ *and* $n_0 > 0$ *such that the following statements hold for all* $n \geq n_0$, *and all* $\epsilon_n$ *satisfying* $2 - c_1 \geq \epsilon_n \geq C_2 / \sqrt{\log n}$ *for all* $n \geq n_0$, *with* $d = (2 - \epsilon_n) n \log n$:

1.  $\mathbf{P}\left[\mathbf{K}_{\backslash i} \textit{ is invertible, } \max_{i \in [m]} \left|y_i \mathbf{y}_{\backslash i}^\mathsf{T} \left(\mathbf{K}_{\backslash i}^{-1} - \frac{1}{d} I_{n-1}\right) \mathbf{X}_{\backslash i} \mathbf{x}_i\right| \leq \frac{\epsilon_n}{2C_0}\right] \geq 1 - \frac{1}{n}.$

2.  $\mathbf{P}\left[\max_{i \in [m]} \left|\frac{1}{d} y_i y_{[m] \backslash i}^\mathsf{T} \mathbf{X}_{[m] \backslash i} \mathbf{x}_i\right| \leq \frac{\epsilon_n}{2C_0}\right] \geq 1 - \frac{1}{n}.$

*Proof.* We start with the first claim. By Lemma 1 and a union bound, we have with probability at least $1 - 1/n$, $\mathbf{K}_{\backslash i}$ is invertible and

$$\max_{i \in [m]} \left|y_i \mathbf{y}_{\backslash i}^\mathsf{T} \left(\mathbf{K}_{\backslash i}^{-1} - \frac{1}{d} I_{n-1}\right) \mathbf{X}_{\backslash i} \mathbf{x}_i\right| \leq C \sqrt{\frac{n \log(mn)}{d}} \left(\sqrt{\frac{n + \log(mn)}{d}} + \frac{n + \log(mn)}{d}\right).$$

For sufficiently large $n$, we have $d \geq c_1 n \log n$ and

$$C\sqrt{\frac{n \log(mn)}{d}} \left( \sqrt{\frac{n + \log(mn)}{d}} + \frac{n + \log(mn)}{d} \right) \leq 2C\sqrt{\frac{n \log(mn)}{d}} \sqrt{\frac{n + \log(mn)}{d}}$$

$$\leq \frac{3Cn\sqrt{\log n}}{(2 - \epsilon_n)n \log n} + \frac{4C\sqrt{n} \log n}{c_1 n \log n}.$$

The second term on the right-hand side is at most $C_2/(4C_0\sqrt{\log n})$ for sufficiently large $n$, and hence at most $\epsilon_n/(4C_0)$. The first term on the right-hand side is also at most $\epsilon_n/(4C_0)$ provided that

$$(2 - \epsilon_n)\epsilon_n \geq \frac{12C_0 C}{\sqrt{\log n}},$$

which is equivalent to

$$\epsilon_n \geq 1 - \sqrt{1 - \frac{12C_0 C}{\sqrt{\log n}}}$$

(since we already assume $\epsilon_n \leq 2 - c_1$ for sufficiently large $n$). This is satisfied provided that $C_2 \geq 12C_0 C$. This proves the first claim.

For the second claim, we have by Lemma 3 (with $n$ in the statement of Lemma 3 set to $m - 1$) and a union bound, we have with probability at least $1 - 1/n$:

$$\max_{i \in [m]} \left| \frac{1}{d} y_i y_{[m] \setminus i}^{\mathsf{T}} \mathbf{X}_{[m] \setminus i} \mathbf{x}_i \right| \leq C \left( \sqrt{\frac{m \log m + m \log(mn)}{d}} + \frac{\sqrt{m} \log(mn)}{d} \right)$$

$$\leq \sqrt{\frac{C'n}{d}} + \frac{C'n}{d \log n}$$

where the second inequality uses $m = n/\log n \leq n/2$ and holds for sufficiently large $n$, with $C' > 0$ an absolute constant. The second term on the right-hand side is at most $C_2/(4C_0\sqrt{\log n})$ for sufficiently large $n$, and hence also at most $\epsilon_n/(4C_0)$. The first term on the right-hand side is also at most $\epsilon_n/(4C_0)$ provided that

$$\frac{16C_0 C'}{\log n} \leq (2 - \epsilon_n)\epsilon_n^2.$$

Since $\epsilon_n \leq 2 - c_1$, the above condition holds as long as $C_2 \geq 4\sqrt{C_0 C'/c_1}$. This proves the second claim. $\square$

**Lemma 6.** *For any $C_0 > 4$ and $c_1 \in (0, 2)$, there exists $C_2 > 0$ such that the following holds for all sequences $(\epsilon_n)$ satisfying $2 - c_1 \geq \epsilon_n \geq C_2/\sqrt{\log n}$ for all large enough $n$, with $d = (2 - \epsilon_n)n \log n$:*

$$\lim_{n \to \infty} \mathbf{P} \left[ \max_{i \in [m]} \frac{1}{d} y_i y_{\setminus [m]}^{\mathsf{T}} \mathbf{X}_{\setminus [m]} \mathbf{x}_i \geq 1 + \frac{\epsilon_n}{C_0} \right] = 1.$$

*Proof.* Observe that conditioned on $\mathbf{X}_{\setminus [m]}$, the $m$ random variables

$$\frac{1}{d} y_i y_{\setminus [m]}^{\mathsf{T}} \mathbf{X}_{\setminus [m]} \mathbf{x}_i, \quad i = 1, \ldots, m$$

are distributed as independent mean-zero Gaussian random variables with variance

$$\sigma^2 := \frac{1}{d^2} \| y_{\setminus [m]}^{\mathsf{T}} \mathbf{X}_{\setminus [m]} \|_2^2.$$

Therefore, the claim is equivalent to

$$\lim_{n \to \infty} \mathbf{P} \left[ \max_{i \in [m]} \sigma \mathbf{g}_i \geq 1 + \frac{\epsilon_n}{C_0} \right] = 1,$$

where $\sigma^2$ is as defined above, and $\mathbf{g}_1, \ldots, \mathbf{g}_m$ are i.i.d. standard Gaussian random variables, independent of $\sigma^2$.

Observe that

$$\mathbf{G} := \frac{1}{\sqrt{n-m}}\mathbf{X}^{\mathsf{T}}_{\backslash[m]} y_{\backslash[m]}$$

is a standard Gaussian random vector in $\mathbb{R}^d$. By Gaussian concentration of Lipschitz functions [26, page 41, 2.35], the following holds with probability at least $1 - 1/n$:

$$
\begin{aligned}
\sigma &\geq \frac{\sqrt{n-m}}{d}\left(\mathbf{E}\left[\|\mathbf{G}\|_2\right] - \sqrt{2\log n}\right) \\
&\geq \frac{\sqrt{n-m}}{d}\left(\sqrt{d} - \frac{1}{2\sqrt{d}} - \sqrt{2\log n}\right) \\
&= \sqrt{\frac{n-m}{d}}\left(1 - \frac{1}{2d} - \sqrt{\frac{2\log n}{d}}\right) \\
&= \frac{1}{\sqrt{2\log n}}\frac{1}{\sqrt{1-\frac{\epsilon_n}{2}}}\sqrt{1 - \frac{1}{\log n}}\left(1 - \frac{1}{2d} - \sqrt{\frac{2\log n}{d}}\right)
\end{aligned}
\tag{15}
$$

where the second inequality follows from standard approximations of the Gamma function, and the final inequality holds assuming $C_2 \geq 1$.

Let $\mathsf{E}$ be the event in which (15) holds. Then

$$
\begin{aligned}
\mathbf{P}\left[\max_{i\in[m]}\sigma\mathbf{g}_i \geq 1 + \frac{\epsilon_n}{C_0}\right] &\geq \mathbf{P}\left[\max_{i\in[m]}\sigma\mathbf{g}_i \geq 1 + \frac{\epsilon_n}{C_0}\ \Big|\ \mathsf{E}\right]\left(1 - \frac{1}{n}\right) \\
&\geq \mathbf{P}\left[\max_{i\in[m]}\mathbf{g}_i \geq \alpha_n\sqrt{2\log n}\right]\left(1 - \frac{1}{n}\right)
\end{aligned}
$$

where

$$
\alpha_n := \frac{\left(1 + \frac{\epsilon_n}{C_0}\right)\sqrt{1 - \frac{\epsilon_n}{2}}}{\sqrt{1 - \frac{1}{\log n}}\left(1 - \frac{1}{2d} - \sqrt{\frac{2\log n}{d}}\right)}.
$$

We claim that the probability on the right-hand side tends to 1 with $n \to \infty$ as well.

The distribution of the random variable $\max_{i\in[m]}\mathbf{g}_i$ obeys a limiting Gumbel distribution; specifically, for all $x > 0$,

$$\lim_{m\to\infty}\mathbf{P}\left[\max_{i\in[m]}\mathbf{g}_i \geq \sqrt{2\log m} - \frac{x + C\log\log m}{\sqrt{\log m}}\right] = 1 - e^{-e^x}$$

where $C > 0$ is an absolute constant [18]. Therefore, it suffices to show that for all $x > 0$, we have

$$\sqrt{2\log m} - \frac{x + C\log\log m}{\sqrt{\log m}} - \alpha_n\sqrt{2\log n} \geq 0$$

for all sufficiently large $n$. Dividing through by $\sqrt{2\log n}$ and using $m = n/\log n$, the above inequality is implied by

$$\sqrt{1 - \frac{\log\log n}{\log n}} - \frac{x + C\log\log(n/2)}{\log n}\frac{1}{\sqrt{1 - \frac{\log\log n}{\log n}}} - \alpha_n \geq 0.\tag{16}$$

Since $0 \leq \epsilon_n \leq 2 - c_1$, we have

$$\left(1 + \frac{\epsilon_n}{C_0}\right)\sqrt{1 - \frac{\epsilon_n}{2}} \leq 1 - \frac{C_0 - 4}{2C_0\sqrt{2c_1}}\cdot\epsilon_n$$

by a Taylor series argument. So, (16) is implied by

$$\frac{C_0 - 4}{2C_0\sqrt{2c_1}}\cdot\epsilon_n - T(n) \geq 0$$

where

$$T(n) = \left( \frac{\log \log n}{\log n} - \frac{x + C \log \log(n/2)}{\log n} \left( 1 + \frac{\log \log n}{\log n} \right) \right)$$

$$\cdot \left( \sqrt{1 - \frac{1}{\log n}} \left( 1 - \frac{1}{2d} - \sqrt{\frac{2 \log n}{d}} \right) \right).$$

Since $\epsilon_n \geq C_2/\sqrt{\log n}$ and $T(n) = o(1/\sqrt{\log n})$, we can choose $C_2$ large enough so that (16) holds for all sufficiently large $n$. $\square$

We conclude as in the proof of Theorem 3. The event in which all of the following hold has probability approaching 1 as $n \to \infty$ by combining Lemma 5 and Lemma 6 and a union bound:

1. $\max_{i \in [m]} \left| y_i y_{\backslash i}^{\mathsf{T}} \left( \mathbf{K}_{\backslash i}^{-1} - \frac{1}{d} I_{n-1} \right) \mathbf{X}_{\backslash i} \mathbf{x}_i \right| \leq \frac{\epsilon_n}{2C_0}$;

2. $\max_{i \in [m]} \left| \frac{1}{d} y_i y_{[m] \backslash i}^{\mathsf{T}} \mathbf{X}_{[m] \backslash i} \mathbf{x}_i \right| \leq \frac{\epsilon_n}{2C_0}$;

3. $\max_{i \in [m]} \frac{1}{d} y_i y_{\backslash [m]}^{\mathsf{T}} \mathbf{X}_{\backslash [m]} \mathbf{x}_i \geq 1 + \frac{\epsilon_n}{C_0}$.

In this event, there exists $i \in [m]$ such that

$$y_i y_{\backslash i}^{\mathsf{T}} \left( \mathbf{K}_{\backslash i}^{-1} - \frac{1}{d} I_{n-1} \right) \mathbf{X}_{\backslash i} \mathbf{x}_i + \frac{1}{d} y_i y_{[m] \backslash i}^{\mathsf{T}} \mathbf{X}_{[m] \backslash i} \mathbf{x}_i + \frac{1}{d} y_i y_{\backslash [m]}^{\mathsf{T}} \mathbf{X}_{\backslash [m]} \mathbf{x}_i$$

$$\geq -\frac{\epsilon_n}{2C_0} - \frac{\epsilon_n}{2C_0} + 1 + \frac{\epsilon_n}{C_0} = 1.$$

### C.2 Above the threshold

Now we consider the case where the dimension is above the threshold, specifically $d = (2 + \epsilon_n) n \log n$.

By Proposition 1, it suffices to show that

$$\lim_{n \to \infty} \mathbf{P} \left[ \exists i \in [n] \text{ such that } y_i y_{\backslash i}^{\mathsf{T}} \mathbf{K}_{\backslash i}^{-1} \mathbf{X}_{\backslash i} \mathbf{x}_i \geq 1 \right] = 0.$$

This is implied by the following lemma combined with a union bound over all $i \in [n]$.

**Lemma 7.** *There exists $C_2 > 0$ and $n_0 > 0$ such that the following statement holds for all $n \geq n_0$, and all $\epsilon_n$ satisfying $\epsilon_n \geq C_2/\sqrt{\log n}$ for all $n \geq n_0$, with $d = (2 + \epsilon_n) n \log n$:*

$$\text{for each } i \in [n]: \quad \mathbf{P} \left[ y_i y_{\backslash i}^{\mathsf{T}} \mathbf{K}_{\backslash i}^{-1} \mathbf{X}_{\backslash i} \mathbf{x}_i \geq 1 \right] \leq \frac{1}{2n\sqrt{\pi \log n}} + \frac{1}{n^2}.$$

*Proof.* Conditional on $\mathbf{X}_{\backslash i}$ (and the probability 1 event that $\mathbf{X}_{\backslash i}$ has rank $n - 1$), the distribution of

$$y_i y_{\backslash i}^{\mathsf{T}} \mathbf{K}_{\backslash i}^{-1} \mathbf{X}_{\backslash i} \mathbf{x}_i$$

is a mean-zero Gaussian with variance

$$\sigma_i^2 := \| y_{\backslash i}^{\mathsf{T}} \mathbf{K}_{\backslash i}^{-1} \mathbf{X}_{\backslash i} \|_2^2 = y_{\backslash i}^{\mathsf{T}} \mathbf{K}_{\backslash i}^{-1} y_{\backslash i}.$$

By Lemma 2, we have for some absolute constant $C > 0$ and sufficiently large $n$, with probability at least $1 - 1/n^2$,

$$\sigma_i^2 \leq n \cdot \mu_{\max}(\mathbf{K}_{\backslash 1}^{-1})$$

$$\leq \frac{n}{d} \left( 1 + C \left( \sqrt{\frac{n + 2 \log n}{d}} + \frac{n + 2 \log n}{d} \right) \right)$$

$$\leq \frac{1}{(2 + \epsilon_n) \log n} \left( 1 + \frac{C'}{\sqrt{\log n}} \right),$$

where $C' > 0$ is a constant depending only on $C$. Let $\mathsf{E}$ be the aforementioned event. Then

$$\mathbf{P}\left[y_i y_{\backslash i}^\top \mathbf{K}_{\backslash i}^{-1} \mathbf{X}_{\backslash i} \mathbf{x}_i \geq 1\right] \leq \mathbf{P}\left[y_i y_{\backslash i}^\top \mathbf{K}_{\backslash i}^{-1} \mathbf{X}_{\backslash i} \mathbf{x}_i \geq 1 \mid \mathsf{E}\right] + \mathbf{P}\left[\neg \mathsf{E}\right]$$

$$\leq 1 - \Phi\left(\sqrt{\frac{(2+\epsilon_n)\log n}{1 + C'/\sqrt{\log n}}}\right) + \frac{1}{n^2}$$

$$\leq \sqrt{\frac{1}{2\pi} \cdot \frac{1 + C'/\sqrt{\log n}}{(2+\epsilon_n)\log n}} \exp\left(-\frac{1}{2} \cdot \frac{(2+\epsilon_n)\log n}{1 + C'/\sqrt{\log n}}\right) + \frac{1}{n^2} \quad (17)$$

where the final inequality follows by the Mills ratio bound (Fact 1). By letting $C_2 \geq 2C'$, we have

$$\frac{2 + \epsilon_n}{1 + C'/\sqrt{\log n}} \geq 2$$

(since we assume $\epsilon_n \geq C_2/\sqrt{\log n}$), upon which bound in (17) is at most

$$\frac{1}{2n\sqrt{\pi \log n}} + \frac{1}{n^2}$$

as claimed. $\qquad\qquad\qquad\qquad\qquad\qquad\qquad\qquad\qquad\qquad\qquad\qquad\qquad\qquad\qquad\qquad\qquad\square$

## D Supplementary material for Section 5

This appendix gives a refined statistical analysis of our hypothesis that SVP is universal for $\ell_2$-SVMs under the assumption that features are drawn identically and independently. We visually assert this universality with Figures 1 and 5, and formally test our hypothesis using a parametric statistical approach, borrowing several ideas from Donoho and Tanner [16].

As described in Section 5, we show the significance of this universality by providing a parametric model that complies with the given universality hypothesis and fits well to the observed rates of SVP. That is, this model permits slight difference for different sample distributions, as long as this difference decays to zero with $n$. Furthermore, we show that the model becomes statistically insignificant if we incorporate extra parameters to allow non-decaying dependence on sample distributions to conclude universality.

Alongside these universality results, we experimentally support the universality of the bounds on transition width, which are proved for the isotropic Gaussian sample case in Section 4.

These analyses are implemented in Python and R. Our code-base can be found on Github at `https://github.com/scO0rpion/SVM-Proliferation-NIPS2021`.

### D.1 Experimental procedures

We conduct a Monte Carlo simulation in order to validate our theoretical results and grasp their generality to distributions with different tail distributions. For the range $(n, d) \in \{40, 42, \ldots, 100\} \times \{100, 110, \ldots, 1000\}$ we study our problem in the following way:

- We generate features $\mathbf{X} \in \mathbb{R}^{n \times d}$ by drawing each $\mathbf{x}_i$ independently from the suite of distributions shown in Table 1. Subsequently, we generate a balanced set of labels $y \in \{\pm 1\}^n$ where the first $\lfloor \frac{n}{2} \rfloor$ samples are assigned class $+1$ and the rest $-1$.
- We used the quadratic program solver from CVXOPT [42][5] to solve (Interpolation Dual) with $p = q = 2$ to tolerance level $10^{-7}$.
- We deem that SVP occurs for an instance of the simulation if the optimizer's output lies in the interior of $\mathbb{R}_+^n$.
- We report the fraction $\hat{\mathbf{p}}$ of $M$ trials that exhibit SVP. Based on Figure 4, we choose the simulation size value $M = 400$ as an appropriate choice for having small enough variance for our range of $(n, d)$. Throughout this section, we run $M = 400$ simulations unless stated otherwise.

---

[5]CVXOPT is distributed at `https://cvxopt.org/` under a GPLv3 license.

| Name | Support | Mean | Variance | Subgaussian? |
|---|---|---|---|---|
| Uniform | $[-1, 1]$ | 0 | $1/3$ | Yes |
| Bernoulli | $\{0, 1\}$ | $1/2$ | $1/4$ | Yes |
| Rademacher | $\{-1, 1\}$ | 0 | 1 | Yes |
| Laplacian | $\mathbb{R}$ | 0 | 2 | No |
| Gaussian | $\mathbb{R}$ | 0 | 1 | Yes |
| Gaussian Biased | $\mathbb{R}$ | 1 | 1 | Yes |

Table 1: The suite of distributions used in experiments. Features $\mathbf{x}_i \in \mathbb{R}^d$ are drawn from a product distribution $\mathcal{D}^{\otimes d}$ of one of the distributions $\mathcal{D}$ in this table.

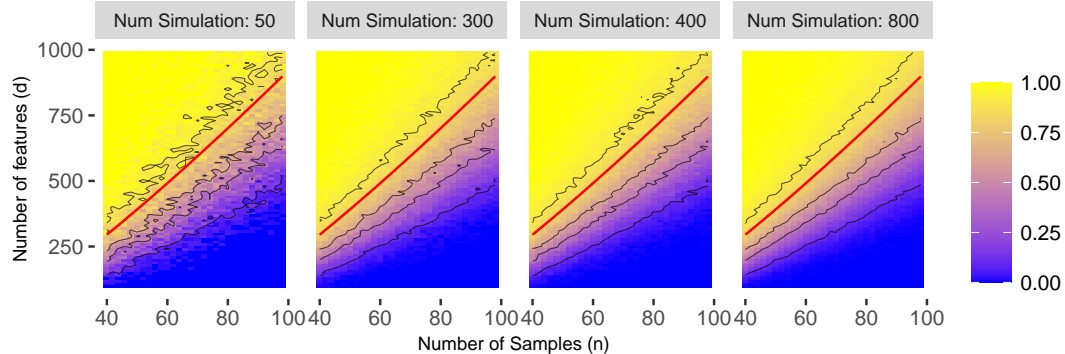

Figure 4: The sensitivity of the experiments to simulation size $M$. The blue curves are 0.1, 0.4, 0.6, and 0.9 quantile contours respectively for the observed rate of SVP for Gaussian samples. The red curve is limiting phase transition boundary, i.e., $n \mapsto 2n \log(n)$

.

Our computing environment was a shared high-performance cluster, where a standard node has two Intel Xeon Gold 6226 2.9 GHz CPUs (each with 16 cores) and 192 GB memory. It took roughly eight hours to run all SVP simulations for $\ell_2$-SVMs on a single node. The simulations for $\ell_1$-SVMs for Figure 3 (Appendix E) took two days to run on the cluster, again just on a single node.

### D.2 Observed universality

Let $\hat{\mathbf{p}} := \hat{\mathbf{p}}(n, d; \mathcal{D}, M) \sim \text{Binom}(p(n, d; \mathcal{D}), M)$ be the observed SVP rate corresponding to a sample distribution $\mathcal{D}$ with independent components and with simulation size $M$. Due to the log-linear dependence of the dimension $d$ of SVP threshold occurs on $n$ from Theorem 4, we parameterize the probability of SVP as a function of $n$ and $\tau := d / (2n \log n)$ instead. The objective here is to provide a reasonable parametric model for $\hat{\mathbf{p}}(n, d; \mathcal{D})$ as an inferential tool to test the universality hypothesis. To do so, we translate our universality hypothesis in the language of our parametric model and ensure that necessary statistical assumptions hold to make inferential claims.

**Model:** We use Probit regression (a generalized linear model with Probit link function) to explain the transition behavior and allow the coefficients to depend explicitly on the distribution $\mathcal{D}$ under which sample components are drawn from. We justify this specific choice of link function at the end of this subsection. We propose the following parametric model; we motivate the terms in the model at the end of the subsection as well.

$$p(n, d; \mathcal{D}) = \Phi\left(\mu^{(0)}(n, \mathcal{D}) + \mu^{(1)}(n, \mathcal{D})\tau + \mu^{(2)}(n, \mathcal{D})\log \tau\right), \tag{18}$$

where

$$\mu^{(i)}(n, \mathcal{D}) = \mu_0^{(i)}(\mathcal{D}) + \frac{\mu_1^{(i)}(\mathcal{D})}{\sqrt{n}},$$

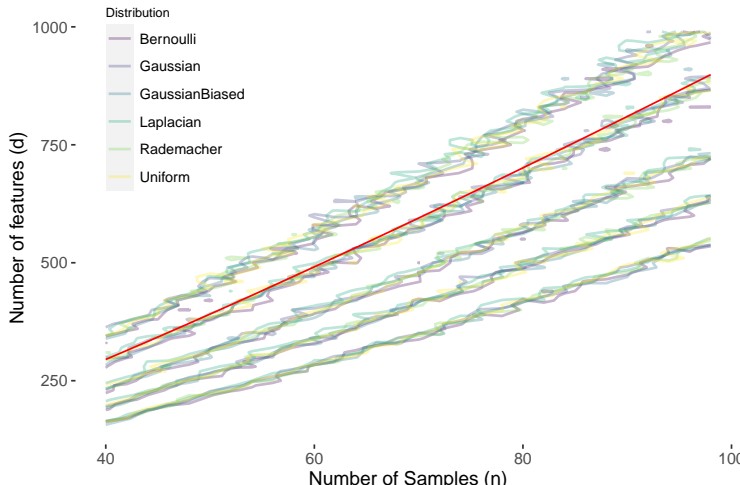

Figure 5: Quantile plots. The observed rates of SVP $\hat{p}$ are similar for all simulated distributions. The quantile plots visualize the dimension $d$ needed for SVP to occur on a 0.2, 0.4, 0.6, 0.8, and 0.9 fraction of the trials for a fixed umber of samples $n$. The red line corresponds to the asymptotic boundary $n \mapsto 2n \log n$, which closely aligns to the level curve corresponding to a 0.8 fraction of trials exhibiting SVP.

and

$$\Phi(t) = \int_{-\infty}^{t} \frac{1}{\sqrt{2\pi}} e^{-x^2/2} \, \mathrm{d}x$$

is the Probit link function (i.e., the standard normal distribution function).

The universality hypothesis can be translated to a testing framework under the model described (18) by prohibiting $\mu_0^{(i)}(\mathcal{D})$ from depending on the underlying distribution $\mathcal{D}$.

- **Universality Hypothesis:** $\mu_0^{(1)}(\mathcal{D})$ are identical for each distribution $\mathcal{D}$, and $\mu_0^{(1)}(\mathcal{D}) = \mu_0^{(2)}(\mathcal{D}) = 0$.
- **Alternative Hypothesis:** $\mu_0^{(i)}(\mathcal{D})$ depends on the underlying distribution $\mathcal{D}$ for some $i \in \{0, 1, 2\}$.

The Universality Hypothesis permits differences from the ground mean (which must decay to zero as $n$ grows large), but requires that other terms be identical. The Alternative Hypothesis instead permits non-decaying difference among distributions. We show that our model **does not reject** the Universality Hypothesis.

**Inference:** We perform Probit regressions on observed SVP rates $\hat{p}$ sequentially on three different models, each of which is a sub-model of its successor. The second and third models correspond to the Universality Hypothesis and the Alternative Hypothesis respectively. We compare their goodness-of-fit using analysis of deviance (ANOVA) to assess whether each subsequent model restriction meaningfully improves on its predecessor's ability to fit the data [22].

- **Model 1:** Does not allow any deviations for different distributions; i.e., $\mu^{(i)}(n, \mathcal{D})$ does not depend on $\mathcal{D}$ for $i \in \{0, 1, 2\}$.
- **Model 2:** Allows deviations in bias that decay to zero; i.e., only $\mu_1^{(i)}(n, \mathcal{D})$ may vary with $\mathcal{D}$.
- **Model 3:** Full model described in Equation (18).

Based on this sequential test given in Table 2, we find that **Model 1** should be rejected, because **Model 2** substantially improves on it in a statistically significant manner. Furthermore, no statistical significance were detected for rejecting **Model 2**, and nearly all the excessive parameters (for **Model 3**) were statistically insignificant. Therefore, we accept **Model 2**, which complies with the Universality Hypothesis.

| Models Compared | Degrees of Freedom | Deviance | P-Value |
|:---:|:---:|:---:|:---:|
| 1 vs. 2 | 5 | 1695.46 | $2 \cdot 10^{-16}$ |
| 2 vs. 3 | 25 | 29.25 | 0.253 |

Table 2: Analysis of deviance for the sequence of three Probit models. P-Values are computed based on Chi-squared tails.

Assuming that the Probit model is well-specified (e.g., the residuals satisfy the usual assumptions), we conclude that **Model 2** is significant. The remainder of the section argues that the Probit model is the most appropriate for this setting. A full analysis of all three fitted models, including the fitted model parameters and p-values for these parameters, can be found in the code repository.

**Motivating the model:** The proposed model (18) is supported by a series of empirical observations about the SVP rate $\hat{\mathbf{p}}$:

- $\hat{\mathbf{p}}$ increases as $\tau$ increases and is mostly unaffected as $n$ changes (left panel (a) of Figure 2). Therefore, the dependence on $n$ should be negligible.
- $\hat{\mathbf{p}}$ is asymmetric around the theoretical boundary $\tau = 1$ (left panel (a) of Figure 2). This behavior motivates the non-linear term in our proposed model and likely originates from the asymmetry of the limiting Gumbel distribution in Section 4.
- As $\tau$ varies, the slope of $n \mapsto p(n, \tau; \mathcal{D})$ changes, which motivates including terms that manage the interaction between $n$ and $\tau$ (right panel (b) in Figure 2).
- As suggested by Donoho and Tanner [16], including a dependence on $1/\sqrt{n}$ is motivated by the theory of Edgeworth expansions, which states that the non-asymptotic behaviour of a random Gaussian problem decays to its asymptotic behavior by some power of the "problem size."

**Model diagnostics:** While our proposed model yields formal evidence of universality, it remains to show that the model is correct, particularly because even wrong models are often statistically significant. To avoid this pitfall, we validate the underlying assumptions required to make inferential claims in our logistic regression model. Figure 6 demonstrates that **Model 2** accurately approximates the observed probabilities $\hat{\mathbf{p}}$ when the probabilities are not too close to one or zero.[6] The figure further shows that the residuals approximately follow a standard normal distribution and do not seem to correlate with the fitted values. Therefore, the residuals corresponding to this model satisfy the usual assumptions necessary for regression.

**Justification for the link function and log:** We test three link functions for **Model 2** to choose an appropriate one for our parametric model (18):

$$\text{Cauchit}(t) = \frac{1}{\pi}\left[\tan^{-1}(t) + \frac{\pi}{2}\right], \qquad \text{Logit}(t) = \frac{1}{1 + e^{-t}}, \qquad \text{Probit}(t) = \Phi(t).$$

We justify the use of $\log \tau$ as a non-linear term in (18) by substituting the logarithmic term with the *Cox-Box transform* of $\tau$, i.e., $(\tau^{\gamma} - 1)/\gamma$ and cross-fitting various link functions with different exponents $\gamma = 0.2, 0.4, 0.5, 1$. The diagnostic plots in Figure 7 indicate that the Probit link function fits best. Moreover, smaller values of $\gamma$ fit observed probabilities better by comparing the deviance r-squares in Table 3 as a measure of goodness of fit. Hence, we use the $\log \tau$, which is the limit of the Cox-Box transform when $\gamma \to 0$.

### D.3 Width of transition

To estimate the width of the phase transaction, we adopt a non-parametric approach. For $q < 1/2$ and fixed $n$ we define the *q-transition zone* to be the range $[\tau_q, \tau_{1-q}]$, where $\tau_q$ and $\tau_{1-q}$ correspond to the ratio $d/(2n\log n)$ for smallest and largest value of $d$ where support vector proliferation occurs with probability $q$ and $1 - q$ respectively. In other words, within this range, the corresponding probabilities are inside $[q, 1 - q]$ interval.

---

[6]These extreme cases are not concerning because we are primarily interested in values of $d$ and $n$ where the asymptotic probabilities are non-degenerate. The significance of **Model 2** continues to hold, even if we restricted attention to a smaller region with non-degenerate SVP rates $\hat{\mathbf{p}}$, such as $\tau \in [0.4, 1.6]$.

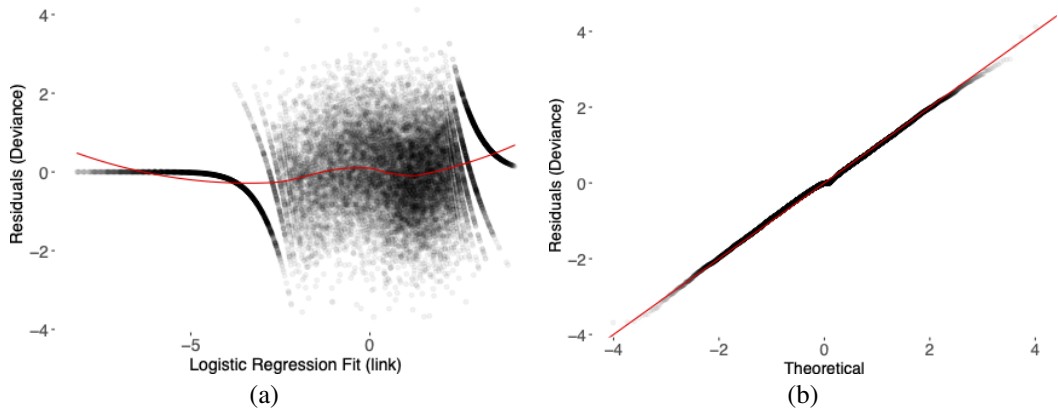

(a)            (b)

Figure 6: Diagnostic plots. *Left panel (a):* The relationship between logistic regression residuals and fitted probabilities for **Model 2** applied to the inverse link function. The red curve is LOESS smoothing to illustrate the trend. The observed probabilities drop to zero faster than tails of Probit function. *Right panel (b):* Quantile plot of the residuals. The red line corresponds to standard normal quantiles.

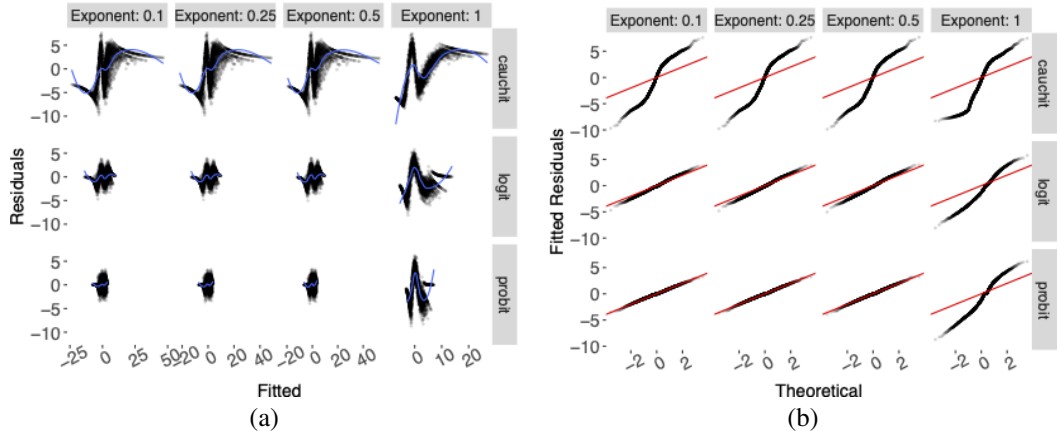

(a)            (b)

Figure 7: Diagnostic plots for Gaussian distribution. *Left panel (a):* The relationships between the fitted values from logistic regression and the residuals for various link functions and exponents. We use LOESS smoothing to visualize the trend and plot it in blue. Asymptotically, the residuals should appear independent of the fitted values; hence, bumpy blue curves indicate non-compliant behaviour with model assumptions. *Right panel (b):* Residual QQ-plots. One expects residuals to asymptotically follow a normal distribution (with a perfect fit corresponding to the overlaid red line), so smaller deviations from a normal distribution is desired.

Formally, we define *scaled q-transition width estimate* as,

$$\hat{w}_q(n,d) = \frac{\hat{\tau}_{1-q} - \hat{\tau}_q}{\Phi^{-1}(1-q) - \Phi^{-1}(q)} \sqrt{\log n},$$

where $\Phi$ is the Probit link function introduced in (18) and $\hat{\tau}_q$ and $\hat{\tau}_{1-q}$ are plug-in estimates of $\tau_q$ and $\tau_{1-q}$. We plot $\hat{w}_q(n,d)$ with $n$ in Figure 8.

## E    Supplementary material for Section 6

### E.1    $\ell_1$ SVM experimental design

To generate Figure 3, we use $M = 400$ Monte Carlo simulations for each pair $(n,d)$ for both $\ell_1$ and $\ell_2$ SVMs with Gaussian ensemble features $\mathbf{X}$ and labels $y$ generated identically to what

|  | | **Link** | | |
|---|---|---|---|---|
| | | **Cauchit** | **Logit** | **Probit** |
| $\gamma$ | **0.10** | 0.9606 | 0.9953 | 0.9971 |
| | **0.25** | 0.9602 | 0.9950 | 0.9969 |
| | **0.50** | 0.9595 | 0.9946 | 0.9967 |
| | **1.00** | 0.9501 | 0.9771 | 0.9722 |

Table 3: $R^2$ values computed from on the fraction between the null deviance and the fitted deviance for different link functions and exponents. The Probit link function with $\gamma = 0.1$ has the best fit.

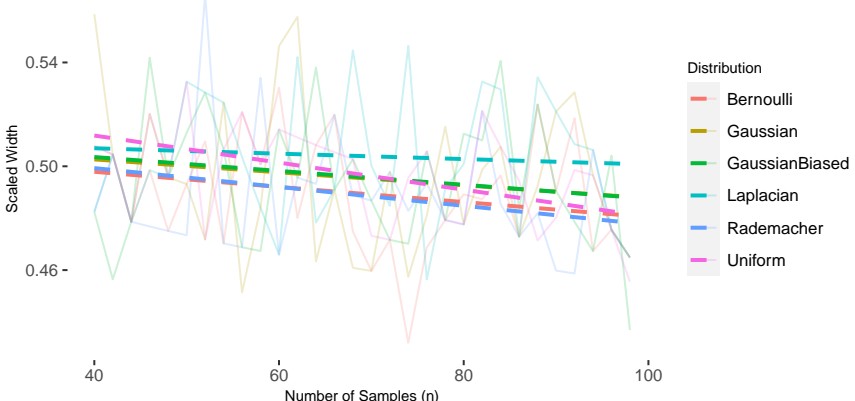

Figure 8: The estimated scaled transition widths for variety of distributions. Note that $\hat{\tau}_q$ and $\hat{\tau}_{1-q}$ are computed with both observed probabilities and non-parametric smoothing spline versions of probabilities which are shown in left and right panels respectively. The overall trend is overlaid using a linear regression applied to the scale width.

discussed in Appendix D. The range of values of $(n, d)$ for $\ell_1$-SVM is within $\{10, 11, \dots, 70\} \times \{80, 85, \dots, 1000\}$. The remainder of the $\ell_1$ experiment is identical to the aforementioned $\ell_2$ experiments. We determine whether SVP occurs for the $\ell_1$ case by solving the (Dual L1) linear program for a $\mathbf{X}$ and $y$ and identifying whether the resulting $\alpha^*$ is in $\mathbb{R}^n_+$. We use the linear program solver in CVXOPT [42] with the default configuration (`absolute tolerance` $= 10^{-7}$, `relative tolerance` $= 10^{-6}$, `feasibility tolerance` $= 10^{-7}$).

### E.2 A geometric interpretation of $\ell_1$ SVM proliferation

A persistent line of work in the statistics and machine learning literature studies phenomena with $\ell_1$ constraints by geometrically analyzing the random mathematical programs. One such application is *exact sparse recovery*, which assumes the existence of a sparse "ground truth" that one aims to recover using linear observations by solving a convex optimization problem. Donoho and Tanner [15] and Amelunxen et al. [2] show the existence of a phase transition in this and similar problems by translating the optimality conditions into a geometrical question as to whether two cones share a ray. Motivated by this line of work, we believe that Conjecture 1 at its core is a geometric phenomenon. Along those lines, we translate the problem of determining whether $\alpha \in \mathbb{R}^n_+$ into a geometric problem that aims to characterize the faces of a random polytope.

By the Fundamental Theorem of Linear Programming, the optimal solution(s) to (Dual L1) lie on corner points of the polytope

$$\mathcal{C}^* = \{\alpha \in \mathbb{R}^n : \|\mathbf{A}^\top \alpha\|_\infty \le 1\}$$

in the dual space. The following lemma provides an alternate characterization the optimal solution $\alpha^*$ to the linear program by relating the corner points of $\mathcal{C}^*$ to the faces of its dual polytope, $\mathcal{C}$.

**Lemma 8.** *Suppose $\mathbf{A} \in \mathbb{R}^{n \times d}$ is a full rank matrix for $d > n$. Then $\alpha^*$ is an optimal solution to* (Dual L1) *iff it is perpendicular to a facet of $\mathcal{C} = \mathrm{Conv}\{\pm \mathbf{A}_{.,i} : i \in [d]\}$ which intersects with the ray passing through origin and $\mathbf{1} \in \mathbb{R}^n$.*

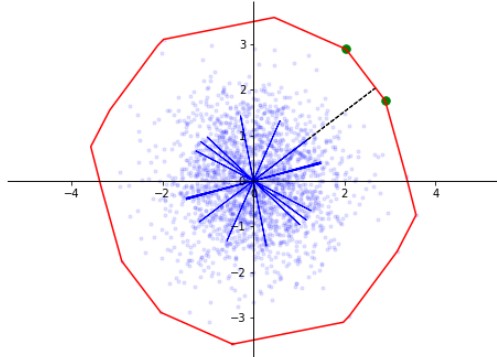

Figure 9: Depiction of the dual space. Blue points correspond to rows of a Gaussian ensemble and their reflections with respect to origin, i.e. $\pm\mathbf{A}_{i,:}$. The outer red convex hull represents $\mathcal{C}$. Blue lines are corner points of $\mathcal{C}^*$, which are perpendicular to the faces of $\mathcal{C}$, and the black line is the direction of the optimal solution $\alpha^*$. Green points indicate the active constraints where the inner product with $\alpha^*$ is equal to one.

As a result of Lemma 8, we can determine whether $\ell_1$ support vector proliferation occurs by instead determining whether the ray in the direction of $\mathbf{1}$ passes through a facet of $\mathcal{C}$ whose closest point to the origin lies in $\mathbb{R}_+^n$. This reduces the problem to understanding the geometry of the facets of random polytope $\mathcal{C}$. If there are sufficiently many facets relative to $2^n$ and most facets cover a very small number of orthants, then it becomes increasingly likely that the facet intersected by the $\mathbf{1}$ ray will also have its projection from $\mathbf{0}$ lie in the positive orthant and support vector proliferation is likely to occur. This intuition can be supported from Dvortzky-Milman Theorem [17] since for a Gaussian ensemble $\mathbf{A}$ one expects the convex hull of its columns to be isomorphic to a sphere when $d = \exp\left(\Omega(n)\right)$. This geometric approach informs our conjecture that the phase transition is likely to occur at the rate $f(n) = \exp\left(\Theta(n)\right)$.

Figure 9 for a Gaussian Ensemble with $n = 2$ and $d = 1500$ and illustrates the relationships between $\mathcal{C}, \mathcal{C}^*, \alpha^*$, and $\mathbf{A}_{\cdot,i}$. The geometric intuition conveyed in the figure is limited by its low value of $n$; we expect qualitatively different behavior to arise when $n$ is large, particularly when considering the geometry of the facets of $\mathcal{C}$. For instance, when $n = 2$, very few samples correspond to corners of the convex hull, and the vast majority lie in the interior. When $n$ is larger, almost all of the samples will be corners of the convex hull, unless $d$ grows exponentially with $n$.

To prove Lemma 8, we make use of several useful facts about the geometric properties of dual polytopes, which are immediate consequences of Farkas' Lemma [see 31].

**Fact 2.** *Let $\mathcal{C} = \mathrm{Conv}\{\pm A_{\cdot,i} : i \in [d]\}$ and $\mathcal{C}^*$ defined as above. Then for a full rank matrix $\mathbf{A} \in \mathbb{R}^{n \times d}$ the following holds:*

- *The origin $\mathbf{0}$ is in the interior of $\mathcal{C}$ and $\mathcal{C}^*$.*
- *$\mathcal{C} = (\mathcal{C}^*)^* := \{\alpha \in \mathbb{R}^n : \langle\alpha', \alpha\rangle \leq 1, \ \forall\alpha' \in \mathcal{C}^*\}$, in words $\mathcal{C}$ and $\mathcal{C}^*$ are polar.*
- *Corner points of $\mathcal{C}^*$ are perpendicular to facets (boundary hyperplanes) of $\mathcal{C}$.*

Now, we are ready to prove Lemma 8.

*Proof of Lemma 8.* We consider the (Dual L1) optimization problem and move constraints into the objective for some Lagrange multiplier $\beta \in \mathbb{R}_+$:

$$\max_{\alpha \in \mathbb{R}^n} \mathbf{1}^\intercal\alpha + \beta\left(1 - \max_{i \leq 2d}\mathbf{B}_i^\intercal\alpha\right)$$

where $\mathbf{B}_i = -\mathbf{B}_{i+d} = \mathbf{A}_{\cdot,i} \in \mathbb{R}^n$ is the $i$th column of $\mathbf{A}$.

If $\alpha^*$ is an optimal solution, then zero must be a subgradient of the objective at $\alpha^*$. Since the set of subgradients of $\alpha \mapsto \max_{i \in [2d]} \mathbf{B}_i^\mathsf{T} \alpha$ at $\alpha$ is $\mathrm{Conv}\{\mathbf{B}_i : i \in I_\alpha\}$, where $I_\alpha = \{i \in [2d] : \langle \mathbf{B}_i, \alpha \rangle = \max_{i' \in [2d]} \langle \mathbf{B}_{i'}, \alpha \rangle)\}$ denotes the set of active constraints at $\alpha$, an optimal solution $\alpha^*$ must satisfy

$$\frac{1}{\beta} \mathbf{1} \in \mathrm{Conv}\{\mathbf{B}_i : i \in I_{\alpha^*}\}.$$

This convex hull is a face of $\mathcal{C}$ since $\langle \mathbf{B}_j, \alpha^* \rangle < \langle \mathbf{B}_i, \alpha^* \rangle = 1$ for any $j \notin I_{\alpha^*}$ and $i \in I_{\alpha^*}$. Moreover, since $\alpha^*$ is also a corner point of $\mathcal{C}^*$ we have $|I_{\alpha^*}| = n$. Combining with Fact 2, we conclude that $\alpha^*$ must be perpendicular to the facet of $\mathcal{C}$ that intersects with the ray passing through $\mathbf{1}$. Existence of such a facet is ensured by the fact that origin is in the interior of $\mathcal{C}$. $\qquad\square$

### E.3 Lower-bound on dimension needed for $\ell_1$ SVP

Based on the previous relationship between support vector proliferation and the faces of a random convex polytope, we can prove a very loose bound on the minimum dimension $d$ needed for support vector proliferation to occur by bounding the number of faces of the polytope. When $d = O(n)$, the polytope will have far fewer than $2^n$ facets, which makes it impossible for most orthants to to be covered by projections from $\mathbf{0}$ onto facets. This (along with rotational invariance of the standard Gaussian distribution) allows us to show that support vector proliferation occurs with a negligibly small probability in this regime.

**Theorem 6.** *Let $\mathbf{A} \in \mathbb{R}^{n \times d}$ be a matrix of i.i.d. $\mathcal{N}(0,1)$ random variables with $d \geq n$. Let $\mathbf{S}$ denote the set of maximizers of* (Dual L1). *If $d < Cn$ for some universal constant $C > 1$, then*

$$\limsup_{n \to \infty} \mathbf{P}\left[\mathbf{S} \cap \mathrm{int}(\mathbb{R}_+^n) \neq \emptyset\right] = 0.$$

A key geometric insight for the proof is supplied by [15]. Let $k^*$ be the maximum integer $k$ such that every $k$ points from among $\{\pm\mathbf{A}_{\cdot,i} : i \in [d]\}$ spans a $k$-dimensional face of $\mathcal{C}$ (i.e., their convex hull is a $k$-dimensional face of $\mathcal{C}$). Then $k^* = \Omega_{\mathbf{P}}\left(\frac{n}{\log(d/n)}\right)$ for large enough values of $n$ and $d$. Hence, it is plausible to expect that every selection of $n$ points spans a facet of $\mathcal{C}$.

*Proof.* Let $F$ denote the facets of $\mathcal{C}$, and to each facet $f \in F$, we associate a corner point $\alpha^{(f)}$ of $\mathcal{C}^*$. Also let $\mathbf{U} \in \mathbb{R}^{n \times n}$ denote a uniformly random rotation matrix, independent of $\mathbf{A}$. Note that the collection $(\alpha^{(f)})_{f \in F}$ will also have a rotational invariant distribution. Since $\mathbf{A}$ has rank $n$ almost surely, Lemma 8 implies that an optimal solution $\alpha^*$ to (Dual L1) must be one of these corner points $\alpha^{(f)}$. Thus, we can upper bound the event that any given optimal solution $\alpha^*$ to (Dual L1) lies in the positive orthant as follows:

$$\mathbf{P}\left[\alpha^* \in \mathrm{int}(\mathbb{R}_+^n)\right] \leq \mathbf{P}\left[\bigcup_{f \in F}\{\alpha^{(f)} \in \mathrm{int}(\mathbb{R}_+^n)\}\right]$$

$$= \mathbf{E}\left[\mathbf{P}\left[\bigcup_{f \in F}\{\mathbf{U}\alpha^{(f)} \subset \mathbb{R}_+^n\}\,\Big|\,(\alpha^{(f)})_{f \in F}\right]\right]$$

$$\leq \mathbf{E}\left[\sum_{f \in F}\mathbf{P}\left[\{\mathbf{U}\alpha^{(f)} \subset \mathbb{R}_+^n\}\,\Big|\,(\alpha^{(f)})_{f \in F}\right]\right]$$

$$\leq \frac{\mathbf{E}\left[|F|\right]}{2^n}.$$

The second inequality is a union bound, and the final inequality uses the fact that $\mathbf{P}[\mathbf{U}\alpha \in \mathbb{R}_+^n] = 1/2^n$ for any fixed $\alpha$. A crude upper bound of $\binom{d}{n}$ on the number of facets gives

$$\mathbf{P}[\alpha^* \in \mathbb{R}_+^n] \leq \frac{\binom{d}{n}}{2^n}.$$

The right-hand side converges to zero as $n \to \infty$ provided that $d < Cn$ for some absolute constant $C \approx 1.29$. $\qquad\square$

Needless to say, the gap between Theorem 6 and the $\ell_1$ support vector proliferation threshold exhibited in Figure 3 is substantial. Indeed, if Theorem 6 were tight, it would imply that support vector proliferation would occur for *smaller* values of $d$ for $\ell_1$ than $\ell_2$, which contradicts our experimental results and geometric intuition. We believe that union bound corresponding to the first inequality in our proof accounts for that looseness. That inequality would be tight only if the existence of some $\alpha^{(f)} \in \mathrm{int}(\mathbb{R}_+^n)$ implies that $\alpha^* \in \mathrm{int}(\mathbb{R}_+^n)$; however, this ignores the possibility that facets span many different orthants and that the facet intersecting the ray through $\mathbf{1}$ may be "close to the origin" in many orthants simultaneously. We believe that a more precise understanding of the geometry of the faces of random high-dimensional polytopes could tighten this bound and hence elucidate the $\ell_1$ support vector proliferation phenomenon.