# OpenReview forum: "Support vector machines and linear regression coincide with very high-dimensional features"
_NeurIPS.cc/2021/Conference — NeurIPS 2021 Poster_

### Official Review · Reviewer_cFdd · 2021-07-07

**Rating:** 7
**Confidence:** 4

**Summary:**

This paper explores the generality of support vector proliferation and makes the following three contributions: (1) proving a super-linear lower bound on the dimension (in terms of sample size) required for support vector proliferation in independent feature models. (2) identifying a sharp phase transition in Gaussian feature models. (3) investigating `the phenomenon of support vector proliferation for $\ell_1$ variant of the SVM.

**Limitations And Societal Impact:**

The potential limitations of this paper from my perspective have been listed in the Main Review above, while this paper has minimal near-term applications with social implications.

**Main Review:**

This paper focuses on the phenomenon of support vector proliferation, where every training example used to fit an SVM becomes a support vector.  Specially, this paper studies the case in which support vector machine (SVM) and ordinary least squares (OLS) return exactly the same hypothesis for sufficiently high-dimensional data. In real-world applications, linear regression models are often used to solve classification problems. This paper theoretically tells us when linear regression model is equivalent to the classification model SVM. Here, I list some problems and suggestions from my perspective:

1. For SVM, this paper only focuses on the hard-margin SVM, however, soft-margin SVM is more commonly-used in real-world applications.

2. As stated in lines 27-28, "When the solution is not unique, it is natural to take the solution of minimum Euclidean norm". In other words, for OLS, this paper only focuses on the case that the number of feature dimension is larger than the number of examples. This conjecture can also be supported by the conclusion obtained in this paper, $d=O(n \log n)$. However, in many real-world applications, the number of feature dimension is smaller than the number of examples. Maybe authors consider the nonlinear mapping feature space with kernel trick?

3. The assumption for the data distribution is too strong, and the theoretical results are not verified over real-world data sets. It is better to add some experiments over some widely-used publicly available data sets.

4.  What are the differences of $\Omega(\cdot)$, $O(\cdot)$, $\Theta(\cdot)$ and $\omega(\cdot)$? For example, line 37: $d = \Omega (n \log n) $, $d = O(n)$, line 54: $d = \Theta (n)$, line 82: $d = \omega (n \log n)$.


#UPDATES AFTER RESPONSE
I have read the responses of authors and other reviews. Although authors did not address my concerns, I still would like to keep my acceptance recommendation as I think that this paper is interesting and an accepted paper does not have to be perfect. Thanks.

**Time Spent Reviewing:**

18

---

> ### Author Response · Authors · 2021-08-09
> **Response to Reviewer cFdd**
>
> We thank the reviewer for their comments and suggestions. Changes will be made accordingly in a revised version of the paper.
>
> Below, we respond to some points raised in the review.
>
> 1. The reviewer writes, "For SVM, this paper only focuses on the hard-margin SVM, however, soft-margin SVM is more commonly-used in real-world applications."
>
> The number of support vectors in soft-margin SVMs has been studied in previous works (e.g., [3,37]), and the characterization depends crucially on non-separable data. It is less clear what happens for separable data. While some prior works (e.g., [7]) have studied the asymptotic limit with $d/n$ approaching a fixed ratio, those results could never exhibit SVP. The lower-bounds in our paper shed light on why that is the case.
>
> In addition, the iterates of gradient descent on the logistic loss objective for linear predictors converge to the hard-margin SVM solution, which makes the hard-margin SVM an object of interest. See, for instance, "The implicit bias of gradient descent on separable data" by Soudry et al (2017).
>
> 2. "[...] the number of feature dimension is smaller than the number of examples. Maybe authors consider the nonlinear mapping feature space with kernel trick?"
>
> In some ML applications, the dimension of the input data will be larger than the number of examples. For example, the feature dimension in genomics data could be regarded as linear or superlinear in $n$. The same goes for many other cases where high-dimensional sparse linear regression is used.
>
> We agree with the reviewer that studying nonlinear mappings is interesting as well, but this is outside the scope of the present paper. It would be difficult to prove rigorous theoretical bounds about these mappings, due to the loss of independence between components. However, we believe the interplay of the numbers of examples and the effective dimensions of nonlinear mappings to be an interesting subject for empirical study. Because Fig. 1 of [23] already explores the role of effective dimensionality for trigonometric features, we opted to focus instead on other distributions for our empirical results. That said, we agree this would be interesting future work.
>
> 3. "The assumption for the data distribution is too strong, and the theoretical results are not verified over real-world data sets."
>
> We agree that the data distributions considered are not realistic. However, our goal is to establish a mathematical result and so we needed a mathematically tractable distribution. We explored some relaxations of the distributional assumptions in the experiments, and the results suggest that the conditions of the theorem could possibly be relaxed.
>
> 4. "What are the differences of Ω(⋅), O(⋅), Θ(⋅) and ω(⋅)?"
>
> We will add this clarification to the notation section. The Wikipedia page for "Big O notation" (a.k.a. Bachmann-Landau notation) gives a fairly detailed description. The textbook "Introduction to Algorithms" by Cormen, Leiserson, Rivest, and Stein (Chapter 3.1) also gives a nice introduction.

---

### Official Review · Reviewer_p8z3 · 2021-07-09

**Rating:** 7
**Confidence:** 3

**Summary:**

This manuscript investigates the phenomena of \emph{Support Vector proliferation} (SVP) [1,2], in which all points in the training set $x_{i}\in\mathbb{R}^{d}$, $i=1,\cdots, n$ become support vectors, i.e. satisfy $y_{i}\hat{w}^{\top}x_{i}=1$, where $\hat{w}\in\mathbb{R}^{d}$ is the SVM estimator and $y_{i}\in\{-1,1\}$ are the training labels. SVP implies that the SVM estimator coincides with interpolating least-norm linear regression, a fact which has been used in many recent works to study overparametrisation in generalised linear models. In particular, the authors focus in a data model defined by fixed labels $y_{i}$ and sub-Gaussian features $x_{i}$ with zero mean and diagonal covariance $\lambda_{k}$, $k=1, \cdots, d$ (i.e. the labels are independent of the features). The key results are:

- Different equivalent characterisations of SVP for generic $\ell_{p}$, $1< p< \infty$ norm SVM (proposition 1).

- Establishing a lower bound showing that one needs at least $d=n\log{n}$ for SVP to occur in the anisotropic model defined above and $\ell_{2}$-SVM  (Theorem 3). This result tightens the a previous bound from [2] (Theorem 1).

- Establishing a sharp transition at $d = 2n\log{n}$ for SVP in the particular case of isotropic Gaussian data $x_{i}\sim\mathcal{N}(0,I_{d})$, also for $\ell_{2}$-SVM (Theorem 4).

The authors also provide numerical simulations corroborating the theorems, and suggesting that their results apply beyond the theorem's assumptions. Finally, they also provide a conjecture for the equivalent result in the $\ell_1$-SVM case.


[1] Vidya Muthukumar, Adhyyan Narang, Vignesh Subramanian, Mikhail Belkin, Daniel Hsu, and Anant Sahai. *Classification vs regression in overparameterized regimes: Does the loss function matter?*, 2020.

[2]: Daniel Hsu, Vidya Muthukumar, and Ji Xu. *On the proliferation of support vectors in high dimensions*.  In Twenty-Fourth International Conference on Artificial Intelligence and Statistics, 391 2021.

**Ethical Concerns:**

There are no ethical issues.

**Limitations And Societal Impact:**

The authors are clear about some of the limitations of their results. However, I missed a deeper discussion of two limitations:

- SVP is defined in section 2 for $\ell_{p}$-SVM, but Theorems 3 \& 4 focus on the $p=2$ case. Section $6$ addresses the $p=1$ case, but I missed a discussion on the $p>2$ case. Are they qualitatively similar to $p=2$ or significantly different? What are the bottlenecks in the proofs for $p>2$? Did the authors explore these cases numerically (akin to Figs. 1 and 3)?
- The work mostly focus on a data model where the labels are independent from the features. The authors dismiss this limitation in a one sentence comment (lines 200-201). However, at least \emph{a priori}, the definition of SVP seems to crucially depends on the interplay between the labels and the features through the definition of the margins $y_{i}\hat{w}^{\top}x_{i}$. Have the authors explored this claim (numerically or analytically) in models in which the labels depend on the features, e.g. in a 2-Gaussian mixture? I believe this deserves an extended discussion.

Finally, one question: do the results require $n$ to be large or are valid down to $n=O(1)$?

**Main Review:**

**Originality:** This work is a follow-up of [2]. However, it employs alternative proof techniques to improve over their results, which also give a geometrical picture of SVP.

**Quality:** I am not an expert in the proof techniques employed, and therefore I cannot confidently judge the details of the proof. However, the results are sound and the authors provide numerical evidence beyond the theorems scope.

**Clarity**: The manuscript is well organised and the reading flow is smooth. The definitions and assumptions used are clearly stated, and the authors provide intuition where it fits, as well as the main ideas of the proof technique employed. I have two suggestions that I believe could further improve the readability for a non-specialist reader:

- The authors make an extensive use of the $O, o, \Omega, \omega$ notation, without previously defining it. While this is standard, I believe that including a recap of the definition in the notation section could help the reader.
- It would be good to explicitly state that different from Proposition 1, Theorem 3 \& 4 are for the $\ell_{2}$-SVM in the "Outline of our results" section.

**Significance**: SVP is an interesting phenomena occurring in overparametrised problems and which has drawn significant attention recently. Therefore, elucidating when and how it occurs in simple statistical models is, in my opinion, of interest to the NeurIPS community. This works gives a step forward in this direction, sharpening previous findings and providing an alternative picture of the mechanism behind SVP.


**Time Spent Reviewing:**

4h

---

> ### Author Response · Authors · 2021-08-09
> **Response to Reviewer p8z3**
>
> We appreciate the reviewer for pointing out helpful suggestions in order to clarify the paper. Changes will be made accordingly in a revised version of the paper.
>
> Below, we address some points raised in the review.
>
> 1. The reviewer writes, "[...] SVP is defined in section 2 for $\ell_p$-SVM, but Theorems 3 & 4 focus on the $p = 2$ case. Section 6 addresses the $p = 1$ case, but I missed a discussion on the $p > 2$ case. Are they qualitatively similar to $p = 2$ or significantly different? What are the bottlenecks in the proofs for $p > 2$? Did the authors explore these cases numerically (akin to Figs. 1 and 3)?"
>
> We didn't investigate $p > 2$, or really $p$ other than 1 or 2, in any detail. We focused on $p=1,2$ because they are textbook examples of regularizers used for linear models in practice. In future work, it would be of interest to explore other cases on a deeper level both numerically and theoretically, where we may not be able to leverage inner product structure or LP duality.
>
> 2. The reviewer writes, "[...] the definition of SVP seems to crucially depends on the interplay between the labels and the features through the definition of the margins $y w^T x$ . Have the authors explored this claim (numerically or analytically) in models in ii which the labels depend on the features, e.g. in a 2-Gaussian mixture? I believe this deserves an extended discussion."
>
> We agree with the reviewer's point made on labels being chosen independently from features. From a theoretical point of view, more general upper-bounds have been explored in previous works (see [23]). In addition, our lower-bounds holds even under some dependence (e.g., $y = \text{sign}(w^T x)$). However, we thought to exhibit the lower-bound with fixed labels since the argument was simpler. On the experimental side, we did perform some experiments with $y = \text{sign}(w^T x)$ as well which had almost identical conclusions, though only the fixed label case was included for consistency.
>
> 3. The reviewer writes, "[...] do the results require $n$ to be large or are valid down to $n = O(1)$?"
>
> We refer the reviewer to the conditions in Theorem 3: $n$ should be at least a quantity that depends on $\delta$ (desired probability for which SVP occurs).

---

### Official Review · Reviewer_Cct8 · 2021-07-13

**Rating:** 6
**Confidence:** 3

**Summary:**

This paper investigates the phenomenon of support vector proliferation (SVP), that is, when linear classification with minimum-$\ell_p$-norm hard-margin SVM and minimum-$\ell_p$-norm interpolation yield the same solution. A previous paper showed for $p=2$ that SVP occurs with high probability for certain distributions with (effective) input dimension $d = \Omega(n \log n)$, but not for $d = O(n)$. The authors close this gap by showing that SVP does not occur for $d = o(n \log n)$. The authors further specify a more precise bound for standard Gaussian inputs and provide empirical evidence showing similar behavior for other distributions. Finally, the authors present some other theoretical and empirical results to illuminate the behavior for different $p$.

**Limitations And Societal Impact:**

Limitations of the paper are addressed. Potential societal impact is not addressed since this is a theoretical paper.

**Main Review:**

Originality: The novel contributions of the paper are clearly stated. The proofs of the main theorems rely on a novel decomposition where, despite taking a maximum over non-independent terms, one part of the decomposition now consists of independent terms where the distribution of the maximum can be bounded more precisely.

Quality: Claims in the paper are supported by proofs and numerical experiments. The theoretical results look reasonable, although I did not read all of the proofs and the parts that I did read were sometimes not very detailed and left a few questions open (see below).

Clarity: The submission is clearly written. However, I would suggest writing more about the implications of this work, i.e., how this work is relevant to ML (see below).

Significance: The authors provide a much tighter characterization of the occurence of SVP for a certain class of distributions than previous work. Although the considered problem is mathematically interesting, I only see limited significance due to several reasons:
- The improvement is only by a factor $O(\log n)$
- It is unclear to me why a practitioner should care whether SVP occurs or not since it does not seem to have direct implications on generalization.
- The analysis of over-parameterized regime is usually motivated by neural networks, which can behave like linear models sometimes, but usually are not trained using hinge loss.
- Raw input data in ML rarely has dimension $\Omega(n)$. The setting considered in this paper is therefore most interesting when applied to data that is generated using a feature map with high-dimensional feature space, or when considering a kernel method with potentially infinite-dimensional feature space. However, the assumption that the data is generated from distributions with independent components excludes realistic feature maps. I understand that treating such cases theoretically is very hard and likely not possible with a similar proof technique. However, I would suggest extending the experimental results with distributions arising from feature maps, which would also make the hypothesized universality of the $d = 2n\log n$ boundary more convincing. While the formulated results only apply to finite-dimensional data, the formulation with effective dimensions appears to generalize to an infinite-dimensional limit, and although the independent components assumption is not satisfied for typical kernels, it would be interesting to study the effective dimension of a kernel's feature distribution depending on the kernel width or the input dimension, in order to see when SVP might arise in a kernel setting. This could be done by replacing the vector $\lambda$ with the eigenvalues of the kernel's integral operator with respect to a given input dimension. In certain cases, these eigenvalues can be computed precisely, see e.g. Minh et al. for the Gaussian kernel on the sphere (https://link.springer.com/content/pdf/10.1007/11776420_14.pdf). In order to let the effective dimension grow with $n$, I guess either the kernel width should decrease at a certain rate with $n$ or the input dimension should increase with $n$. Asymptotics of the latter type have also been investigated e.g. by Liang et al. (https://arxiv.org/abs/1808.00387, https://arxiv.org/abs/1908.10292).

Perhaps the significance of the paper could also be improved by establishing a link to the Double Descent literature, as the considered setting is exactly in the regime where the Double Descent literature would suggest bad generalization performance of the considered methods. A distinction between overfitting and interpolating classifiers is also made by Belkin et al. (http://proceedings.mlr.press/v80/belkin18a.html), maybe this could be of interest.

Conclusion: The paper contains a clear theoretical contribution to a rather specialized topic of research. Due to concerns about significance, I am currently opting for a weak reject.

Major comments:

L. 545 f.: The matrix $A^\top A$ is a $d \times d$ matrix of rank $\leq n$. In the regime $d > n$, it cannot be invertible. Maybe the shown argument works with pseudoinverses, but this is not immediately clear. Also, the last step in the proof where the authors go back from A to X and y is unclear - is this some sort of Schur complement / Woodbury identity? Maybe this part of the proof could be simplified if the solution of the dual problem can be expressed in terms of the (explicitly known) solution of the primal problem.
Since the corresponding part of the proof is only an alternative proof to an already known statement, this problem does not question the validity of the rest of the paper.

Eq. after line 562: The first inequality appears to have a term of order n \log(n/\delta)^{1/3} and I don't see how this is covered by the third assumption in (3). Maybe it works by inserting the definition of m and using some bound between d_\infty and d_2. In any case, I would recommend adding more details to these arguments, also regarding some of the inequalities after l. 570 (which seem to be correct, though).

Minor comments:

L. 103: "primarily relevant for non-separable data" -> why would that be? Even when using a high-dimensional feature space, regularization might be necessary to avoid overfitting (even if unregularized models are trendy).

L. 137: . missing

L. 138: "matrix" -> "the matrix", "a matrix" or "", similar in L. 136

L. 151: "with high probability as soon as $d = \Omega(n)$" -> In fact, this holds almost surely as soon as $d \geq n$. Proof: The samples are almost surely linearly independent (this holds for all continuous distributions and also for some non-continuous ones, see e.g. Section 5 in https://arxiv.org/abs/2010.01851), hence the interpolation problem is almost surely solvable and hence they are linearly separable. (However, for general $\lambda$-anisotropic subgaussian distributions as considered later in the paper, which are not necessarily continuous, these properties do not necessarily hold almost surely.)

L. 195: How restrictive is it to assume that a random variable is 1-subgaussian with second moment = 1? While almost every subgaussian random variable can be rescaled to have second moment = 1, the rescaled version need not be 1-subgaussian.

L. 207: probability is not exactly 1-\delta, I assume? Also, does this mean that the constant in the \Omega is arbitrary or is it "given" by the theorem?

L. 208: meaning of formulation "with constant probability" is unclear

L. 216 / L. 196: Since the $\ell_2$-case is rotationally invariant, it seems that general covariance matrices could be used and not only diagonal ones. Maybe this could be remarked somewhere near Theorem 3.

L. 260: You use both $\log$ and $\ln$ for the same thing (maybe also in other places). It would be better to consistently use one of the two.

Fig. 1: "Radamacher" -> "Rademacher". Maybe it would be advantageous to use log-log plots to show more clearly that the relative width of the phase transition gets smaller for larger $n$? (Although this effect is separately demonstrated in the appendix.) Maybe it would be helpful to extend the plots a bit to the left, such that the curvature from the $\log(n)$ term is better visible.

Fig. 2: Repeat $d = \tau n \log n$ for the reader? L. 292 says $\tau = d/(2n\log n)$, which is different from $d = \tau n \log n$ used before. The usage of $\tau$ should be made consistent.

Fig. 2: "from the Gaussian samples" -> "using a Gaussian distribution"?

L. 291f.: Notation for model parameters is unintuitive, although it makes sense. Maybe it would be more intuitive to put n and \calD as indices, otherwise \mu looks like a function.

L. 490f.: The principal minors are not the K_{\i} matrices. Also, the invertibility of K ist not equivalent to the invertibility of all K_{\i} as can be seen from the symmetric matrix K = [[1, 1], [1, 1]]. However, it is known here that K ist symmetric and positive semidefinite, and the assumption that K is invertible implies that K is positive definite, which in turn implies that all K_{\i} are positive definite and hence invertible.

L. 491: "invertiability" -> "invertibility"

L. 492: "inveritble" -> "invertible"

The equation after L. 536 is wrong - $\alpha^*$ belongs in a different place. However, the following steps already use the correct variant of the equation, so it is not a big problem.

L. 544: "problem" can be removed

L. 584 and following: Which matrix norm is used? You might want to add an index to the norm to indicate which norm it is - probably the 2-norm.

Equation after l. 588: The last line should have parentheses after $\exp$.

L. 610: Would it be more appropriate to use $n-m$ instead of $n$ here?

**Time Spent Reviewing:**

10

---

> ### Author Response · Authors · 2021-08-09
> **Response to Reviewer Cct8**
>
> We thank the reviewer for their very thorough comments and suggestions. Changes will be made accordingly in a revised version of the paper.
>
> Below, we respond to some points raised in the review.
>
> 1. The reviewer writes that significance of the paper is limited because the "improvement is only by a factor $O(\log n)$."
>
> We don't think this is a reasonable criticism of the results. Removing asymptotically large gaps between upper and lower bounds is commonplace in mathematical research, and we gained new insight into the phenomenon by finding the proofs to remove the gap.
>
> 2. "It is unclear to me why a practitioner should care whether SVP occurs or not since it does not seem to have direct implications on generalization."
>
> An analysis of generalization performance of SVM, obtained using exactly this coincidence between SVM and OLS, was recently conducted by Muthukumar et al (2020), as we discussed in Section 1.2. Please also see our response to Reviewer FQDN regarding this.
>
> 3. "The analysis of over-parameterized regime is usually motivated by neural networks, which can behave like linear models sometimes, but usually are not trained using hinge loss."
>
> The iterates of gradient descent on the logistic loss objective for linear predictors converge to the hard-margin SVM solution, which makes the hard-margin SVM an object of interest even if one does not directly use hinge loss. See, for instance, "The implicit bias of gradient descent on separable data" by Soudry et al (2017). We will address this concern in the introduction of a revision of this paper.
>
> 4. "Raw input data in ML rarely has dimension $\Omega(n)$"
>
> It's not clear how the reviewer means to characterize "raw input data in ML", but it seems to be at odds with real examples from ML applications. For example, the feature dimension in genomics data could be regarded as linear or superlinear in $n$. The same goes for many other cases where high-dimensional sparse linear regression is used.
>
> We agree that examining kernel settings is interesting as well, and we'll refer to the works suggested by the reviewer in a revision. However, this is outside the scope of the paper; we believe the paper stands on its own without this extension.
>
> 5. "[...] I would suggest extending the experimental results with distributions arising from feature maps."
>
> We agree with the reviewer that it would be interesting to empirically study when SVP occurs on distributions with feature maps with different effective dimensions. As the reviewer mentions, dependence between features will make theoretical analysis difficult, but we expect the interplay of $n$ and the analogues of $d_2$ and $d_\infty$ to be illustrative empirically. Because Fig. 1 of [23] already explores the role of effective dimensionality for trigonometric features, we opted to focus instead on other distributions for our empirical results. However, we agree this would be interesting future work.
>
> 6. "The significance of the paper could also be improved by establishing a link to the Double Descent literature [...]"
>
> As we discuss above, Muthukumar et al (2020) uses this exact OLS=SVM connection to establish generalization bounds on SVMs and to directly connect this to the Double Descent literature. Please refer to our response to Reviewer FQDN. We will highlight this connection in a revision of the paper.
>
> 7. "L. 545 f."
>
> This was a typo. The correct quantity to optimize following line 545 is $(A A^T)^{-1/2} \nu^*$. This matrix will then be $n \times n$ and hence invertible. We thank the reviewer for noticing this.
>
> 8. "Eq. after line 562."
>
> We'll clarify how the inequalities in Equation (3) imply the necessary inequalities for Lemmas 1, 3, and 4. The argument for how this particular inequality follows from (3) is as follows:
>
>   - By the definition of $m$, $n \log^{1/3} \frac{3m}{\delta} + n^{1/3} \log \frac{3m}{\delta} \leq n (\frac{d_2}{C_2 n} + \log \frac{3}{\delta})^{1/3} + n^{1/3}(\frac{d_2}{C_2 n} +  \log \frac{3}{\delta}) \leq \frac{n^{2/3}d_2^{1/3}}{C_2}  + n \log^{1/3} \frac{3}{\delta} + \frac{d_2}{C_2 n^{2/3}} + n^{1/3} \log \frac{3}{\delta}$.
>
>   - The first term is at most $\frac{d_\infty}{4}$ by inequalities (3d) and (3c) for large enough $C_3$.
>
>   - The second and fourth terms are at most $\frac{d_\infty}{4}$ by inequality (3c).
>
>   - The third term is at most $\frac{d_\infty}{4}$ by inequalities (3b) and (3a).
>
> We thank the reviewer for the numerous helpful minor comments, including those not specifically addressed below. They are greatly appreciated and we'll implement the suggestions in the revision of the paper.
>
> 9. "L. 103."
>
> The reviewer makes a good point, and we will change the final sentence of that paragraph to remove the inaccurate and irrelevant comment: "In these cases, the asymptotic number of support vectors is shown to be related to the noise rate in the problem. The setups we study are linearly separable, which makes it possible to study the hard-margin SVM (without regularization)."
>
> 10. "L. 151."
>
> This is correct, and we'll clarify the statement. The original statement from Cover is a slightly stronger one, since the $d = \Omega(n)$ condition actually applies for $d \geq \frac{n}{2}$. However, this case is not important for our analysis, and your reference is more clear for the settings we consider.
>
> 11. "L. 195."
>
> The reviewer is correct that the two conditions (1-subgaussianity and unit second moment) need not coincide. The analysis could be adapted easily to to hold for arbitrary $\gamma$-subgaussianity, but we opted not to do so for the sake of simplicity and to avoid having another constant in our theorem statement. However, this could be included for completeness.
>
> 12. "L. 207: [...] does this mean that the constant in the \Omega is arbitrary or is it "given" by the theorem?"
>
> All constants suppressed in $O(\cdot)$, $\Omega(\cdot)$, etc. are universal constants. For example, Theorem 1 could be read as: "There is a universal constant $C>0$ such that the following holds. [...] If $d_\infty \geq C n \log(n) \log(1/\delta)$, [...]".
>
> 13. "L. 490f."
>
> $K_{\setminus i}$ denotes a principal minor of $K$; see the definition on line 137. However, the reviewer is correct in that we only mean to say that positive definiteness of $K$ implies that all principal minors of $K$ are invertible (rather than "equivalence"). We'll correct this in the revision.
>
> 14. "L. 584."
>
> Yes, it is the spectral norm. We will clarify this in the revision.

---

> > ### Comment · Reviewer_Cct8 · 2021-08-23
> > **Response to authors**
> >
> > 1. On one hand, the size of the gap influences how many practical/theoretical scenarios it applies to, and on the other hand, as the authors note, it does not necessarily say anything about the importance of the mathematical insights. Therefore, I would count this as a minor limitation of significance.
> >
> > 2. I agree that my statement was formulated too strongly. As the authors note, SVP can be helpful for proofs, although it is unclear whether it affects generalization. The paper implies that SVP cannot be used for a proof in the regime $d = o(n \log n)$. I would still argue that the impact on generalization analysis is rather small, although there are other aspects in which the paper is more significant.
> >
> > 3. Thanks, this definitely helps.
> >
> > 4. This is a good point, I assume this is also a motivation for you to investigate the $\ell_1$ case. For the $\ell_2$ case, the positive generalization results in [33] apply to an anisotropic model where the important features (i.e. the normal vector of the decision boundary) lie in a direction of high variance. I would guess that this is more realistic for feature-space models than for, e.g., genomics data.
> >
> > By the way, the linked articles should only be seen as suggestions and need not be cited if they don't fit the story of the paper.
> >
> > 5. I am somewhat concerned that using only i.i.d. distributions with random labels in Figure 1 makes the empirical evidence for the hypothesized universality of the threshold $d = 2n\log(n)$ weaker than claimed in the text. As Fig. 1 of [23] only shows two examples, it doesn't really help. I noticed that the hypothesis only applies to isotropic distributions as it is not clear which notion of effective dimension should be used for anisotropic distributions. However, even when only considering isotropic input distributions, I think it would still strengthen your hypothesis (or show up its limitations) to include experiments in Figure 1 with
> > * isotropic non-i.i.d. distributions, and
> > * distributions with input-dependent label distribution. (You mentioned to Reviewer p8z3 that you already conducted such an experiment, if space is problematic you could also put it in the appendix.)
> > For the first point, you could use a uniform distribution on a sphere and/or features $x = (\cos(2\pi k t))_{k=0}^{d-1}, t \sim \mathcal{U}[0, 1]$.
> >
> > Since I am under the impression that the inclusion of such experiments would not be very time-consuming, I would advocate to include them.
> >
> >
> > **Other comments:**
> >
> > * As mentioned by reviewer p8z3, the assumption with an input-independent label distribution is quite unrealistic. As you mention that the proofs can also be generalized to $y = \mathrm{sign}(w^T x)$, would it be feasible to at least put a remark in the appendix on what would need to be changed in the proofs? Regarding your remark in L. 200: Again, the result by Cover can be replaced as in "L. 151" and the replacement works for any assignment of labels.
> >
> > * "L. 195.": I think it should at least be mentioned that the theorem also holds in this case.
> >
> > * "L. 490f.": Good. The usage of "principal minor" is correct, I confused it with "leading principal minor", which would have been wrong.
> >
> > * Maybe it would make sense to draw some isolines in the L1 case in Figure 3 as you have already done in Figure 4?
> >
> >
> > **Conclusion:** Some of my concerns on significance have been addressed by the authors' response. As a result, I raise my score from 5 to 6.

---

> > > ### Author Response · Authors · 2021-09-08
> > > **Thank you!**
> > >
> > > Thank you for your reply and suggestions! We will take them into account in the final version of the paper.

---

### Official Review · Reviewer_FQDN · 2021-07-16

**Rating:** 5
**Confidence:** 3

**Summary:**

This paper proves a super-linear lower bound on the dimension of support vector proliferation (SVP) when features are anisotropic subgaussians. The exact asymptotic threshold of the phase transition in the isotropic Gaussian distribution is also proved. Finally, this paper discussed the case of SVP phase transition for $\ell_1$-SVMs. The experiments on synthetic data sets are also conducted.

**Limitations And Societal Impact:**

The authors are suggested to elaborate more on in which scenarios the SVP theory can be applied?

**Main Review:**

From my point of view, the coincidence between SVM and OLS is not the essential of the problem. If I understand correctly, the paper just wants to show that when dimension is appropriately high and the features are anisotropic subgaussians, the examples will lie on two parallel hyperplanes. Obviously, this is independent of what learning models are chosen.

From the practical perspective, SVP only occurs on a quite idealized condition, thus cannot provide any insight for learning algorithms’ understanding and design. From the theoretical perspective, the techniques used in the proof are just subgaussian concentration bounds as well as some linear algebra. It seems difficult to judge the contributions of the paper.

**Time Spent Reviewing:**

48

---

> ### Author Response · Authors · 2021-08-09
> **Response to Reviewer FQDN**
>
> We thank the reviewer for their comments and suggestions. Changes will be made accordingly in a revised version of the paper.
>
> Below, we respond to some points raised in the review.
>
> 1. The reviewer writes, "[...] the paper just wants to show that when dimension is appropriately high and the features are anisotropic subgaussians, the examples will lie on two parallel hyperplanes. Obviously, this is independent of what learning models are chosen."
>
> This isn't quite accurate, since the main technical contribution of our paper is to prove that the SVP phenomenon *doesn't* happen unless $d / (n \log n)$ is sufficiently large, under the data models that we consider. (There is also the detail about the examples being labeled that adds another small wrinkle to the inaccuracy of the reviewer's remark.)
>
> While the SVP phenomenon can be viewed solely in geometric terms, we opted to present our results about it in the context of SVM and OLS because that connection has proved useful in studying generalization properties of SVM. See Section 1.2 of the paper, as well as our elaboration below.
>
> 2. The reviewer writes, "[...] the techniques used in the proof are just subgaussian concentration bounds as well as some linear algebra. It seems difficult to judge the contributions of the paper."
>
> We think the reductive comment about the proof techniques isn't a  reasonable criticism of the results or the paper itself.
>
> The contributions of the paper can be judged by comparing the prior and related works (e.g., see pages 2-3 of the paper).
>
> 3. "The authors are suggested to elaborate more on in which scenarios the SVP theory can be applied?"
>
> As we discussed in Section 1.2, the coincidence between SVM and OLS was already used in recent works to analyze linear classification in very high dimensions under different data distributions. For example, in [33], the authors studied a data model inspired by the spiked covariance model of Wang and Fan (2017), and they identified a regime of overparameterization where the hard-margin SVM classifier has good generalization behavior (i.e., classification risk going to 0) even when all training examples are support vectors. (Their result was strengthened by an improved characterization of SVP in [23].) Our new lower-bound (Theorem 3) can be viewed as establishing a limit on this approach to the analysis of SVM; specifically, if the (effective) dimension is not sufficiently large, the OLS and SVM solutions may not coincide.
>
> We'll include this elaboration in a revision of the paper.

---

> ### Comment · Reviewer_FQDN · 2021-08-28
> **Raise score to 6**
>
> I have read the response and the authors partially answered my questions, thus I raise my score to 6.

---

> > ### Author Response · Authors · 2021-09-08
> > **Thank you!**
> >
> > Thank you for your reply! Is it possible to adjust your 'rating' score in the actual review?

---

### Official Review · Reviewer_AB4x · 2021-07-23

**Rating:** 7
**Confidence:** 3

**Summary:**

The paper considers the conditions under which the solution to the classic, hard-margin SVM classifier collapses to the solution of the ordinary linear regression problem.  This is known as support vector proliferation and occurs when all training vectors for the SVM are support vectors.  SVP has been observed previously, and an upper bound placed on the dimension (relative to training set size) required to induce it given.  The paper further elaborates on these results, providing improved lower bounds on the dimension for which it occurs.  The transition region in which SVP occurs is characterised and it's width bounded.  Finally, the authors speculate as to the conditions required for SVP in the $\ell_1$ variant of the SVM, and show experimentally that SVP appears to require much higher dimension for this case.

**Limitations And Societal Impact:**

Yes.

**Main Review:**

The results in the paper are technically interesting, if not entirely surprising, and do have some interest in gaining a deeper understanding of the behaviour of the SVM in the limits where current results are not applicable (i.e. when the number of support vectors is large, contradicting the key assumptions of certain results regarding e.g. convergence analysis).  The experimental results that are given back-up the results, at least for small training set sizes.

I do worry that the results are a little on the incremental side, and the hypothesis for the $\ell_1$-SVM is underwhelming (at least to me), but nevertheless overall I would err on the side of acceptance.

POST-REBUTTAL: having read the response and the ensuing discussion I have chosen to leave my response unchanged and recommend accepting this paper.

**Time Spent Reviewing:**

4

---

> ### Author Response · Authors · 2021-08-09
> **Response to Reviewer AB4x**
>
> We thank the reviewer for their comments.
>
> We believe the results will be of interest and can inspire follow-up works on related questions.

---

### Decision · Program_Chairs · 2021-09-27

**Decision:**

Accept (Poster)

**Comment:**

The paper improves existing bounds for the support vector proliferation regime.
While initially there were some concerns among the reviewers, the authors managed
to convince the more skeptical reviewers during the rebuttal phase. On the negative
side, the paper deals, as its predecessors, with a rather restricted data model and direct
practical implications remain vague.